**Letter**

# Migrating immune cells globally coordinate protrusive forces

Patricia Reis-Rodrigues [1], Mario J. Avellaneda [1], Nikola Canigova [1], Florian Gaertner [1,2], Kari Vaahtomeri [1,3], Michael Riedl[1], Ingrid de Vries[1], Jack Merrin[1], Robert Hauschild [1], Yoshinori Fukui [4], Alba Juanes Garcia[1] & Michael Sixt [1] ✉

Efficient immune responses rely on the capacity of leukocytes to traverse diverse and complex tissues. To meet such changing environmental conditions, leukocytes usually adopt an ameboid configuration, using their forward-positioned nucleus as a probe to identify and follow the path of least resistance among pre-existing pores. We show that, in dense environments where even the largest pores preclude free passage, leukocytes position their nucleus behind the centrosome and organelles. The local compression imposed on the cell body by its surroundings triggers assembly of a central F-actin pool, located between cell front and nucleus. Central actin pushes outward to transiently dilate a path for organelles and nucleus. Pools of central and front actin are tightly coupled and experimental depletion of the central pool enhances actin accumulation and protrusion formation at the cell front. Although this shifted balance speeds up cells in permissive environments, migration in restrictive environments is impaired, as the unleashed leading edge dissociates from the trapped cell body. Our findings establish an actin regulatory loop that balances path dilation with advancement of the leading edge to maintain cellular coherence.

The composition and geometry of the interstitium can vary substantially between tissue types, and between physiological and inflammatory states, posing physical and biochemical challenges for migrating immune cells. Unlike immune cells, mesenchymal cells form tight adhesive interactions with the environment and use acto-myosin-mediated pulling forces to deform the interstitial matrix[1]. Whenever transient deformation is not sufficient, mesenchymal cells release proteolytic enzymes to digest a path for the cell body[2,3]. This is facilitated by positioning the centrosome and secretory machinery in front of the nucleus to support local delivery of proteases and adhesion molecules[4,5]. In contrast, fast migrating ameboid cells, such as leukocytes, are more opportunistic[6]. Usually, they neither permanently remodel nor tightly adhere to their environment, and position their nucleus forward, followed by the centrosome and organelles[5,7]. This allows them to use the nucleus as a gauge to probe their vicinity, select larger pores over smaller ones and thereby navigate along a path of least resistance[8]. Ameboid and mesenchymal locomotion strategies were long considered cell-intrinsic properties[9]. However, new evidence suggests that, in response to specific environmental parameters such as extreme confinement, inability to proteolyse and lack of adhesive ligands, mesenchymal cells can also adopt ameboid features[10,11]. To what extent ameboid cells can adopt characteristics from mesenchymal migration is less clear[12,13].

Although ameboid and mesenchymal cells operate in quantitatively very different force regimes, both rely on the actin cytoskeleton

[1]Institute of Science and Technology Austria (ISTA), Klosterneuburg, Austria. [2]Department of Medicine I, University Hospital, LMU Munich, Munich, Germany. [3]Translational Cancer Medicine Research Program, University of Helsinki and Wihuri Research Institute, Biomedicum Helsinki, Helsinki, Finland. [4]Department of Immunobiology and Neuroscience, Division of Immunogenetics, Medical Institute of Bioregulation, Kyushu University, Fukuoka, Japan. ✉e-mail: Sixt@ist.ac.at

to generate forces. Pulling forces strictly require substrate-specific adhesions that can be quantified accurately by traction force microscopy[14]. Less is known about pushing forces. These can result from cortical acto-myosin contractility because of the build-up of hydrostatic pressure, as exemplified in cellular blebs[15]. Alternatively, actin can also directly polymerize against, and thereby protrude the plasma membrane, as seen in lamellipodia and filopodia. To what extent polymerization-driven pushing forces are sufficient to displace or deform external obstacles is not firmly established[16]. Pushing forces seem especially relevant for ameboid cells that do not transmit strong pulling forces through adhesion receptors.

To ultimately understand how a cell translates intracellular forces into locomotion of the whole cell body it is important to not only study how isolated adhesions or protrusions act on a substrate, but also how mechanical forces are coordinated on the scale of the whole cell. We used mature dendritic cells (DCs) that we derived from immortalized hematopoietic progenitor cells as a model system for highly migratory immune cells[17]. To test whether DCs can adapt their locomotion strategy to environments of differential density, we observed DCs expressing centrin–enhanced green fluorescent protein (eGFP), which labels the microtubule-organizing center (MTOC), directionally migrating towards a CCL19 chemokine gradient in collagen gels of varying concentrations (1.7–3.5 mg ml$^{-1}$) (Fig. 1a). After fixation, we quantified the relative position of the MTOC and nucleus (Hoechst) along the polarization axis. In low density gels, only 20% of DCs migrated MTOC-first, whereas this orientation was observed in 60% of DCs in high density gels (Fig. 1b,c), indicating that DCs can reorient their organelles when encountering narrow pores. As positioning the MTOC in front of the nucleus is typical for mesenchymal cells, which rely on tight adhesions to the substrate, we generated DCs deficient for talin 1 (ref. 18) (Extended Data Fig. 1a,b). $Tln1^{-/-}$ DCs migrated comparably to $Tln1^{+/+}$ in both low and high density collagen gels (Supplementary Video 1 and Extended Data Fig. 1c) and displayed similar percentages of cells migrating MTOC-first (15% and 45%, respectively) (Fig. 1d,e), indicating that organelle reorientation was triggered by geometrical changes, but did not depend on substrate adhesions.

To challenge this finding in controlled geometries, we chemotactically guided DCs expressing the microtubule plus-end protein EB3 labeled with fluorescent mCherry (EB3–mCherry$^+$) through one-dimensional (1D) microfluidic channels with narrow constrictions at the entrance (Fig. 1f). EB3–mCherry$^+$ dynamics showed that virtually all microtubules originated from a single location (Fig. 1f), indicating that, in DCs, the centrosome served as the sole MTOC. As in collagen gels, organelle orientation was dependent on the cross-section of the constriction, with the MTOC-first orientation being more prevalent in

smaller cross-sections (Fig. 1g). When advancing through the straight, unconstricted, part of the channel, cells reverted frequently to a nucleus-first configuration (Fig. 1g and Supplementary Video 2). Upon entering constrictions, DCs coexpressing EB3–mCherry and the actin reporter LifeAct–eGFP consistently showed an intense actin signal inside the constricted area that, as in the MTOC, was located in front of the nucleus (Fig. 1f,h). The intensity of the actin signal increased with decreasing cross-section of the constriction (Fig. 1i,j). By contrast, we observed no obvious actin accumulation in cells migrating through straight channels (Fig. 1f), suggesting that actin accumulation was a response to compression of the cell body. LifeAct–eGFP$^+$ DCs migrating under vertical confinement between two surfaces separated by varying distances (3–8 μm) showed a prominent circular-shaped pool of actin that located in the cell center (Extended Data Fig. 1d and Supplementary Video 2). The number of DCs showing the central actin pool increased with decreasing height of confinement (Extended Data Fig. 1e).

Being confined within stiff environments (such as in the microfluidic setting) and soft environments (such as tissues in vivo, in collagen gels or under layers of soft material), can have different effects on migrating cells. We therefore imaged LifeAct–eGFP$^+$ DCs migrating under soft (0.5%) and stiff (1.0%) agarose. In this set-up, where DCs have to lift the deformable layer of agarose (Fig. 1k), the central actin pool was present in virtually all cells (Extended Data Fig. 1f). To understand whether adhesions are necessary for the formation of the central actin pool, we imaged DCs expressing a GFP-tagged version of the focal adhesion protein VASP (VASP–GFP$^+$) and found that VASP–GFP was absent from the region of the central actin pool (Supplementary Video 3). Moreover, the central actin pool was detectable both in $Tln1^{-/-}$ LifeAct–eGFP$^+$ and wild-type (WT) LifeAct–eGFP$^+$ DCs migrating in passivated substrates (Supplementary Video 3), suggesting that formation of the central actin pool did not require adhesive interactions with the substrate. Under soft agarose, only 50% of DCs showed the central actin pool positioned in front of the nucleus (Fig. 1k,l). The prevalence of this configuration increased to 80–85% under stiff agarose, which was accompanied by a higher intensity of the central actin pool (Fig. 1k–m and Supplementary Video 3). Similarly, we also observed a higher prevalence of MTOC-first migrating DCs in stiff agarose (75%) compared to DCs migrating under soft agarose (50%) (Extended Data Fig. 1g,h and Supplementary Video 3). Other organelles, such as the Golgi apparatus and lysosomes, also positioned in front of the nucleus, close to the central actin pool (Extended Data Fig. 1i). Formation of the central pool of actin was not exclusive to DCs, but was also detected in primary activated T cells isolated from WT LifeAct–eGFP$^+$ mice[19] (Supplementary Video 5 and Extended Data Fig. 1j,k). These observations

**Fig. 1 | Organelle reorientation in DCs. a**, Scheme showing the shape and position of the nucleus in a DC migrating in collagen. **b**, Representative images of centrin–eGFP$^+$ DCs (MTOC, red), labeled with Hoechst in 1.7 and 3.5 mg ml$^{-1}$ collagen matrices. Scale bar, 10 μm. **c**, Percentages of DCs with a MTOC-first orientation in 1.7 mg ml$^{-1}$ collagen ($n = 125$ cells) and 3.5 mg ml$^{-1}$ collagen ($n = 97$ cells) pooled from at least two independent experiments. ****$P < 0.0001$. **d**, Representative images of $Tln1^{+/+}$ and $Tln1^{-/-}$ centrin–eGFP$^+$ DCs (MTOC, red) and labeled with Hoechst in 1.7 and 3.5 mg ml$^{-1}$ collagen matrices. Scale bar, 10 μm. **e**, Percentages of $Tln1^{+/+}$ and $Tln1^{-/-}$ DCs with a MTOC-first orientation in 1.7 mg ml$^{-1}$ collagen ($Tln1^{+/+}$, $n = 42$ cells; $Tln1^{-/-}$ $n = 33$ cells) or 3.5 mg ml$^{-1}$ collagen ($Tln1^{+/+}$, $n = 44$ cells; $Tln1^{-/-}$, $n = 38$ cells); 1.7 mg ml$^{-1}$ collagen, not significant (NS), $P = 0.7640$; 3.5 mg ml$^{-1}$ collagen, NS $P > 0.9999$. **f**, Scheme showing DCs migrating in microfluidic channels (top) and EB3–mCherry$^+$ (MTOC, red, middle) and LifeAct–eGFP$^+$ (actin, black, bottom) DCs labeled with Hoechst (middle and bottom) migrating in channels with constrictions of 6 μm × 2.5 μm, 1.7 μm or 1.2 μm versus straight 6 μm × 6 μm channels. Scale bar, 10 μm. **g**, Percentages of cells showing MTOC-first orientation in straight channels (CH) and channels with constrictions as in **f**. CH, $n = 426$ cells; 2.5 μm, $n = 137$ cells; 1.7 μm, $n = 117$ cells; 1.2 μm, $n = 172$ cells from three independent experiments. $P = 0.5495$ (CH versus 2.5 μm), $P = 0.0566$ (CH versus 1.7 μm) and ****$P < 0.0001$ (CH versus 1.2 μm). **h**, Percentages of cells with actin-

first orientation in channels with constrictions of 2.5 μm ($n = 138$ cells), 1.7 μm ($n = 119$ cells) or 1.2 μm ($n = 162$ cells). Data are pooled from three independent experiments. ****$P < 0.0001$; NS, $P = 0.6405$. **i**, Time-lapse of a LifeAct–eGFP$^+$ (actin, black) DC labeled with Hoechst entering a 1.7 μm constriction (top three rows) and temporal maximum projection of LifeAct–eGFP of the same DC, with the constricted area is highlighted in blue (bottom). Scale bar, 10 μm. a.u., arbitrary units. **j**, Ratios between maximum signal within and outside constrictions of 2.5, 1.7 and 1.2 μm width in single LifeAct–eGFP$^+$ DCs. Data are pooled from three independent experiments; 2.5 μm, $n = 101$ cells; 1.7 μm, $n = 95$ cells; 1.2 μm, $n = 129$ cells. ****$P < 0.0001$; NS, $P = 0.5831$. **k**, Representative images of LifeAct–eGFP$^+$ (actin, black) DCs labeled with Hoechst under 0.5% and 1.0% agarose. Scale bar, 15 μm. **l**, Percentages of DCs with actin-first orientation in 0.5% agarose ($n = 145$ cells) or 1.0% agarose ($n = 88$ cells) pooled from three independent experiments. ****$P < 0.0001$. **m**, Mean intensities of central actin in LifeAct–eGFP$^+$ DCs under agarose, normalized to global actin intensity of the same cell integrated over time. Data are pooled from three independent experiments; 0.5% agarose, $n = 155$ cells; 1.0% agarose, $n = 67$ cells. ****$P < 0.0001$. Hoechst stain in **b**, **d**, **f**, **i** and **k** shows nucleus (blue). Histogram bars in **c**, **e**, **g**, **h** and **l** are mean ± s.e.m. Error bars in **j** and **m** are s.e.m. Two-sided Fisher's exact test (**c**, **e**, **g**, **h**, **l**); two-tailed unpaired Mann–Whitney test (**j**, **m**).

indicated that whenever ameboid migrating immune cells, such as DCs and T cells, transited through narrow spaces, they positioned the MTOC and bulk of organelles in front of the nucleus and assembled a mechanosensitive central pool of actin that responded to physical confinement.

To test whether the mechanoresponsiveness of the central actin pool might counter external forces acting orthogonal to the direction of migration, we developed pushing force microscopy. We incorporated fluorescent beads into agarose and tracked bead displacement using kymographic analysis of fast confocal microscopy stacks (Fig. 2a, Extended Data Fig. 2a and Supplementary Video 6). In the absence of cells, beads remained stationary over time. In contrast, when DCs migrated below them, beads were displaced vertically, indicating that DCs transiently deformed the agarose (Fig. 2b). To locate more

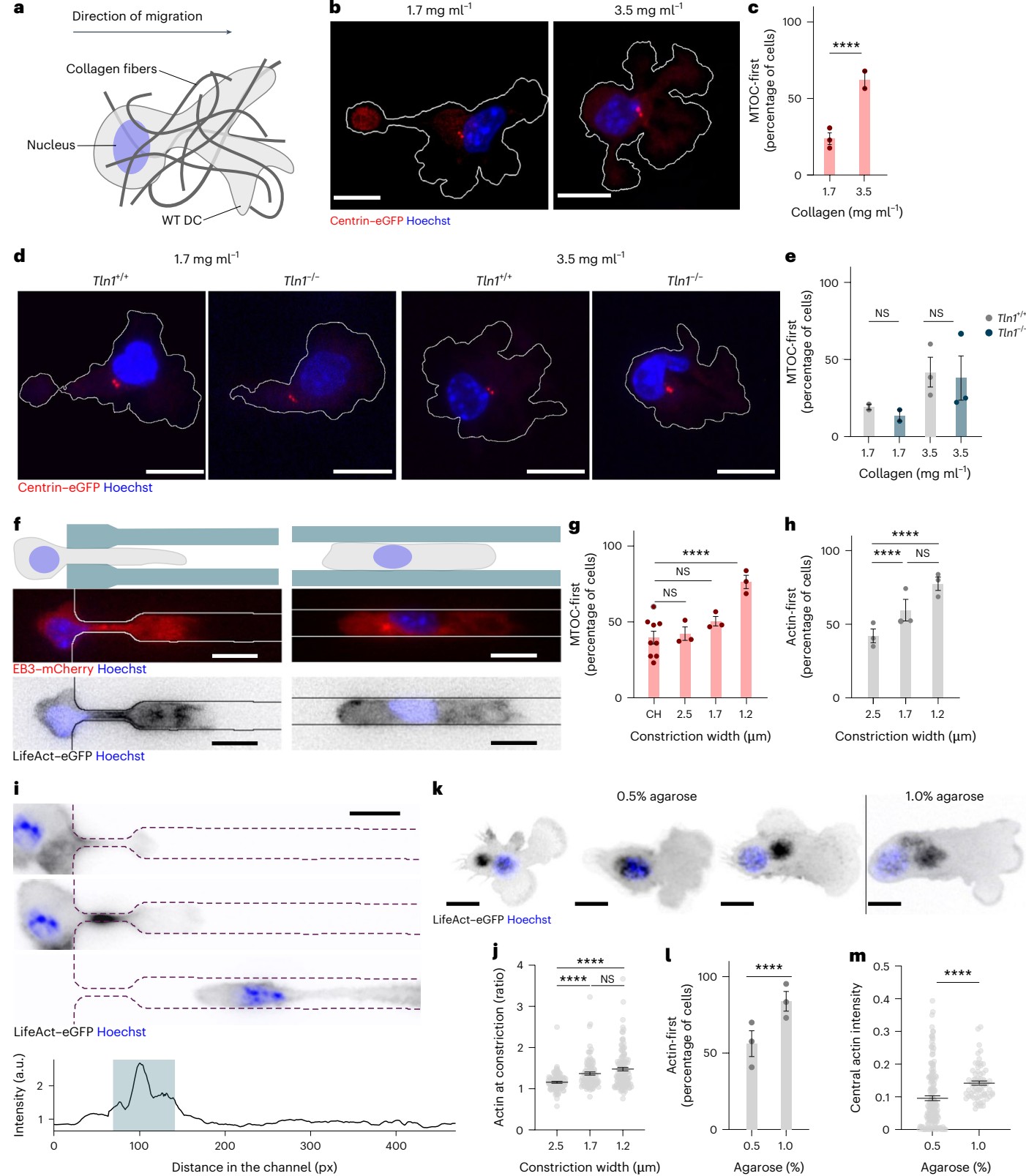

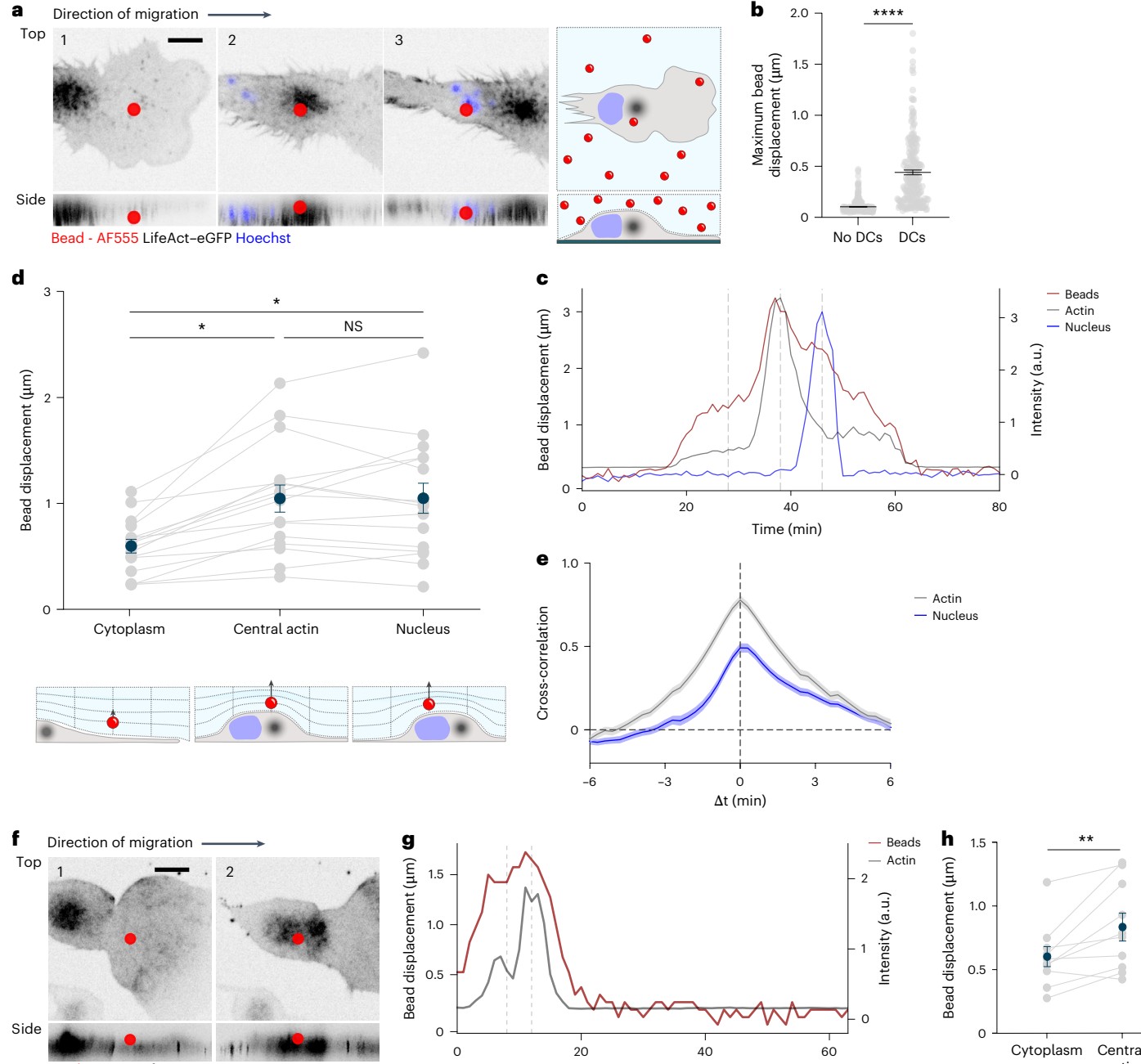

**Fig. 2 | The central actin pool induces substrate deformations. a**, Top view (top) and lateral projection (bottom) of a LifeAct–eGFP⁺ DC (actin, black) labeled with Hoechst (nucleus, blue) migrating under agarose with fluorescent beads labeled with AF555 showing the cell body under the bead (1), the central actin cloud under the bead (2), the nucleus under the bead (3) (left) and a scheme showing a migrating DC under agarose with fluorescent beads (right). Scale bar, 10 µm. **b**, Maximum bead displacement in the absence (no DCs; n = 333 beads) or presence (DCs; n = 205 beads) of LifeAct–eGFP⁺ DCs migrating under agarose. ****P < 0.0001. Two-tailed unpaired Mann–Whitney test. **c**, Change in bead displacement in Z-stacks (beads) and intensities of LifeAct–eGFP (actin) and Hoechst (nucleus) in LifeAct–eGFP⁺ DCs over 80 min. Dashed gray lines highlight the three timepoints shown in **a**. **d**, Contribution of cytoplasmatic, central actin pool and nuclear actin to bead displacement in Z-stacks in LifeAct–eGFP⁺ DCs (n = 16 beads) pooled from at least three independent experiments (top); and scheme showing bead displacement inflicted by LifeAct–eGFP⁺ DCs migrating under agarose (bottom). Gray lines connect measurements in the same bead. Dark blue dots show the mean displacement in Z-stacks. *P = 0.0241 (cytoplasm versus central actin); *P = 0.0230 (cytoplasm versus nucleus); NS, P = 0.9998 (central actin versus nucleus) (one-way ANOVA). **e**, Temporal cross-correlation between bead displacement and nucleus or actin intensities in LifeAct–eGFP⁺ DCs migrating under agarose with AF555⁺ beads. **f**, Top view (top) and lateral projection (bottom) of enucleated LifeAct–eGFP⁺ DCs (actin, black) labeled with Hoechst (nucleus, blue) migrating under agarose with AF555⁺ beads showing the cell body under the bead (1) and the central actin pool under the bead (2). Scale bar, 10 µm. **g**, Change in bead displacement in Z-stacks (beads) and intensity of LifeAct–eGFP (actin) in LifeAct–eGFP⁺ enucleated DCs over 60 min. Dashed gray lines highlight the two timepoints shown in **f**. **h**, Contribution of cytoplasmic and central actin pool to bead displacement in Z-stacks in LifeAc–eGFP⁺ DCs (n = 10 beads) pooled from three independent experiments. **P = 0.083. Two-tailed paired t-test. Error bars in **b**, **d**, **e** and **h** are s.e.m.

precisely which cell components contributed to these deformations, we simultaneously imaged the nucleus (Hoechst) and actin (Life-Act–eGFP), while probing bead displacement. Although beads were detectably displaced by the whole cell body, including the periphery, displacement was most prominent above the central actin pool (Fig. 2c and Extended Data Fig. 2b). Passage of the nucleus sustained the deformations induced by the central actin pool before the substrate relaxed to its original position during nuclear exit (Fig. 2d). Cross-correlation analysis between bead displacement and either LifeAct–eGFP or Hoechst signals indicated that bead displacement was correlated strongly with the presence of actin, whereas the nucleus showed a weaker and asymmetric correlation (Fig. 2e). Similar bead displacements induced by the central actin pool were observed in enucleated DCs (Fig. 2f–h and Supplementary Video 6), indicating that the ability of the central actin pool to deform the substrate did not depend on the nucleus. We detected similar local substrate deformations associated with actin bursts in DCs migrating through collagen I matrices (Extended Data Fig. 2c–e and Supplementary Video 7). Spatial maps of collagen fiber deformation and actin intensity maxima showed that local maxima of fiber displacements were in close proximity to peaks in LifeAct–eGFP signal (Extended Data Fig. 2f,g). These findings supported the notion that cells encountering confined spaces resorted to actin polymerization to locally deform the extracellular environment.

Next, we wondered how perturbations of the central actin pool would affect the ability of cells to migrate and interact with the substrate. Actin dynamics is controlled by the small Rho GTPases Rac1 and Cdc42. Rac1 mainly triggers actin polymerization at the tip of the lamellipodia through direct interaction with the WAVE complex, which in turn activates Arp2/3 dependent nucleation of new branched filaments, whereas Cdc42 has more pleiotropic effects on cytoskeletal dynamics and cell polarity[20,21]. To probe whether and how these pathways affected the central actin pool, while avoiding deranging the homeostasis of cell shape and membrane dynamics, we treated WT DCs migrating under agarose with low concentrations of Rac1 and Cdc42 inhibitors (Extended Data Fig. 3a–c and Supplementary Video 8). Although the Rac1 inhibitor NSC23766 did not have an obvious effect on the percentage of DCs showing a detectable central actin pool (Extended Data Fig. 3d,e), Cdc42 perturbation using either ZCL278 or ML141 inhibitors led to a two-fold decrease in the prevalence of the central actin pool (Fig. 3a,b and Extended Data Fig. 3f,g) and a reduction of the local F-actin signal in the central pool in comparison to untreated cells (Fig. 3c and Extended Data Fig. 3h). No changes in the total amount of cellular F-actin were observed (Fig. 3d). Moreover, WT DCs transiently transfected with a dominant negative mutant of Cdc42 (Cdc42[T17N]–GFP) showed a similar decrease of the percentage of cells with a detectable central actin pool (Extended Data Fig. 3i,j), confirming the key regulatory role of Cdc42.

Among other effectors, Cdc42 triggers WASp-dependent Arp2/3 activation. Analysis of DCs expressing WASp–GFP migrating under agarose showed that WASp–GFP localized not only at the lamellipodium, but also at the region of the central actin pool[22] (Supplementary Video 9). In line with a role for WASp in formation of the central actin pool, the number of *Wasp*[-/-] DCs with a phalloidin positive central actin pool was reduced to 55% (Extended Data Fig. 3k,l).

Cdc42 interacts with several guanine exchange factors, among them DOCK8, which is expressed prominently in the hematopoietic lineage and causative for a severe congenital immunodeficiency associated with actin dysregulation[23–25]. In suspension, *Dock8*[-/-] DCs were morphologically indistinguishable from WT DCs, in line with previous studies[26]. However, when confined under stiff agarose, phalloidin staining revealed a complete lack of the central actin pool (Fig. 3e,f). Notably, WASp–GFP localization at the cell center was also lost, with WASp–GFP dots detectable only at the cellular periphery (Supplementary Video 9). Re-expression of DOCK8–GFP in *Dock8*[-/-] DCs was sufficient to rescue the WT phenotype, and showed that DOCK8–GFP

colocalized with the central actin pool but was not present anywhere else throughout the cell, including the leading edge (Fig. 3g,h). DOCK8–GFP also accumulated at the constriction of microfluidic channels (Fig. 3i,j and Supplementary Video 10). Thus, DOCK8 localization at the center of the cell triggered activation of Cdc42 and recruitment of WASp. Although actin polymerization triggered by WASp contributed to the formation of the central actin pool, it was not essential, suggesting the participation of other Cdc42 effectors.

To investigate the role of the central pool of actin in cell motility, we imaged the chemotactic migration of *Dock8*[-/-] DCs confined under agarose. The migration speed of *Dock8*[-/-] DCs was not different compared to WT DCs (Fig. 4a). In pushing force microscopy, *Dock8*[-/-] DCs inflicted smaller actin-mediated deformations on the agarose (Fig. 4b,c and Supplementary Video 11), with the nucleus being the main bearer of the load in the absence of the central actin pool (Fig. 4d and Extended Data Fig. 4a). In addition, *Dock8*[-/-] DCs displayed distinct elongated morphology and an incoherent leading edge that often branched into two or more lobes (Fig. 4e and Supplementary Video 12). Phalloidin staining of fixed *Dock8*[-/-] DCs migrating under agarose revealed that, although the amount of global F-actin was minimally reduced compared to WT DCs (Extended Data Fig. 4b,c), the F-actin signal at the leading edge was substantially enhanced (Figs. 3e and 4f and Extended Data Fig. 4d). Our results suggest that the lack of the central actin pool in *Dock8*[-/-] DCs impaired substrate deformation and was compensated by a hyperstabilized leading edge, which resulted in jamming of the cell body.

To better understand the communication between actin at the cell front and actin at the cell body, we imaged the dynamics of these two pools in WT LifeAct–eGFP[+] DCs migrating in polydimethylsiloxane (PDMS) pillar mazes, where small obstacles in the migratory path promote splitting of the lamellipodium (Extended Data Fig. 4e). We observed that lamellipodium retractions were accompanied by an increase of LifeAct–eGFP intensity at the central pool (Extended Data Fig. 4f, g and Supplementary Video 12), while no substantial signal variations were detected in other areas of the cell body (Extended Data Fig. 4g). Similarly, we observed a strong negative correlation between actin intensity at protrusion sites and in the central pool in actin–eGFP[+] DCs migrating under agarose (Fig. 4g,h and Supplementary Video 12), suggesting that the two actin pools were strongly coupled and might compete for actin polymerization. To assess quantitatively how this coupling corresponds to migratory dynamics, we performed cross-correlation analysis and found that LifeAct–eGFP intensity at the central pool was correlated negatively with both the projected cell area and cell speed (Fig. 4i, Extended Data Fig. 4h,i and Supplementary Video 12). The positive correlation between cell speed and the projected area was lost in *Dock8*[-/-] DCs (Fig. 4j). These observations suggested that cells redistributed actin between the leading edge and the central pool of actin on demand. Under conditions in which the cell body was largely unobstructed (that is, in the absence of tight constrictions), actin was enriched at the cell front, enhancing leading edge protrusion and accelerating forward locomotion; when cells faced more constrictive environments, actin was recruited to the central pool (Fig. 4k). This dual use allows actin polymerization to dilate a path for organelles and nucleus, preventing cell entrapment in areas of high confinement, and serves as a 'capacitor' by restricting actin accumulation at the cell front, preventing leading edge advancement whenever the cell body is trapped.

Next, we tested whether dysregulation of the central actin pool had a different impact on cell migration depending on the geometry and complexity of the environment. *Dock8*[-/-] DCs migrating in collagen gels were substantially slower than WT DCs (Fig. 5a,b and Supplementary Video 13), as previously reported[26–28], and showed signs of enhanced leading edge stabilization as indicated by the formation of several simultaneous protrusions (Fig. 5c). We also observed a high rate of fragmentation in *Dock8*[-/-] DCs, which often resulted in cell

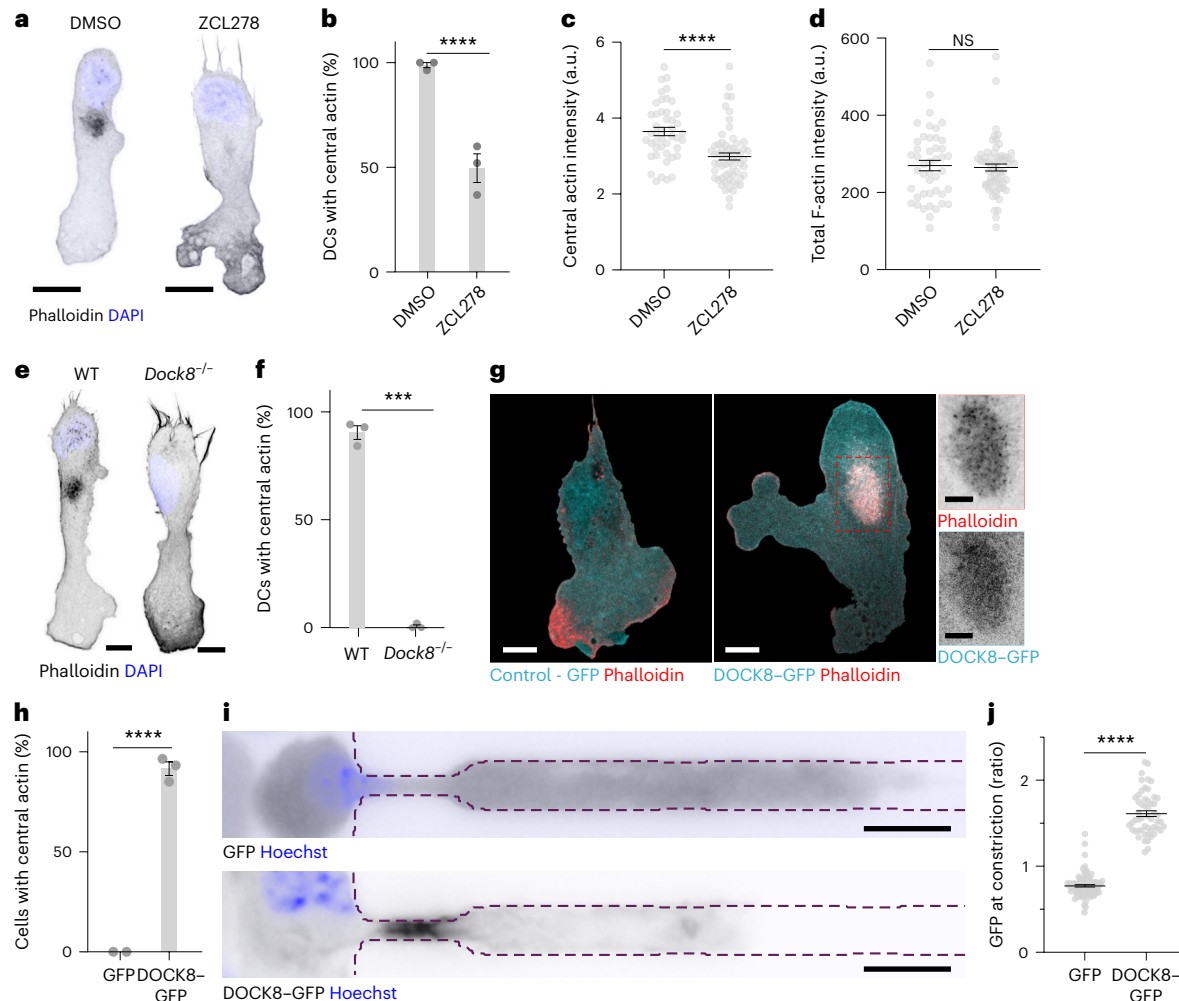

**Fig. 3 | Cdc42 and its exchange factor DOCK8 regulate the central actin pool.**
**a**, Representative images of WT DCs migrating under 1.0% agarose treated with
DMSO or the Cdc42 inhibitor ZCL278 that were fixed and stained for
phalloidin (F-actin, black or red) and DAPI (nucleus, blue). Scale bars, 10 μm.
**b**, Percentages of WT DCs treated with DMSO or ZCL278 with a central actin
pool in three independent experiments. DMSO, $n = 147$ cells; ZCL278, $n = 66$
cells; ****$P < 0.0001$. **c**, Mean central actin pool intensity in WT DCs as in **b**. Mean
intensities were normalized to the global actin intensity in each cell. DMSO,
$n = 45$ cells; ZCL278, $n = 61$ cells; ****$P < 0.0001$. **d**, Mean total F-actin intensity in
WT DCs as in **b**. DMSO $n = 45$ cells; ZCL278, $n = 61$ cells; NS, $P = 0.7455$.
**e**, Representative images of WT and $Dock8^{-/-}$ DCs migrating under 1.0% agarose
treated as in **a**. Scale bars, 10 μm. **f**, Percentages of WT and $Dock8^{-/-}$ DCs migrating
under 1.0% agarose with a central actin pool in three independent experiments.
WT, $n = 240$ cells; $Dock8^{-/-}$, $n = 171$ cells. ****$P < 0.0001$. **g**, Representative images
of $Dock8^{-/-}$ DCs expressing GFP (left) or DOCK8–GFP (right) migrating under

1.0% agarose treated as in **a**. Red dashed box, area used for inset: top, phalloidin
(F-actin, red); bottom, DOCK8–GFP (DOCK8, cyan). Scale bars, 10 μm, 5 μm
(inset). **h**, Percentages of $Dock8^{-/-}$ DCs expressing GFP or DOCK8–GFP showing
a central actin pool during migration under 1.0% agarose treated as in **a**. Data
pooled from two independent experiments. GFP, $n = 47$ cells; DOCK8–GFP, $n = 92$
cells. ****$P < 0.0001$. **i**, Top images of $Dock8^{-/-}$ DCs expressing GFP (top, black) or
DOCK8–GFP (bottom, black) and labeled with Hoechst (nucleus, blue) migrating
in PDMS microchannels with 1.7 μm × 6 μm constriction. Scale bar, 20 μm.
**j**, Ratio between maximum GFP or DOCK8–GFP density at the constriction and
outside of the constriction in GFP+ and DOCK8–GFP+ DCs as in **i**. Data pooled
from three independent experiments. GFP, $n = 91$ cells; DOCK8–GFP, $n = 53$ cells.
****$P < 0.0001$. Histogram bars in **b**, **f**, and **h** are mean ± s.e.m. Error bars in **c**, **d** and
**j** s.e.m. Two-sided Fisher's exact test (**b,f,h**). Two-tailed unpaired Mann–Whitney
test (**c,d,j**).

death (Fig. 5d), consistent with findings in T cells[29]. The cell fragments,
especially those originating from the leading edge, were often motile
and chemotactic (Fig. 5e). $Dock8^{-/-}$ DCs chemotactically migrating in
PDMS mazes with 1-μm to 3-μm-distanced pillars extended several pro-
trusions, entangled and often fragmented (Fig. 5f and Supplementary
Video 13). However, occasionally, $Dock8^{-/-}$ DCs adopted a monopolar
configuration and migrated substantially faster than $Dock8^{-/-}$ DCs
with several competing leading edges (Fig. 5g). To test whether the
enhanced leading edge boosted forward locomotion in simple geom-
etries, in which confinement and leading edge splitting was limited, we
tested the migration of $Dock8^{-/-}$ DCs in straight microfluidic channels.
In this set-up, $Dock8^{-/-}$ DCs migrated substantially faster than WT DCs
(Fig. 5h and Supplementary Video 13). Thus, when DCs were not slowed

down by restrictions imposed by the environment, redistribution of
the central actin pool towards the leading edge enhanced migration.
Here we showed that, when confronted with very narrow constric-
tions, ameboid cells changed their polar configuration by sweeping
the nucleus to the back and positioning the MTOC in front. The local
confinement imposed by the environment also triggered the polymeri-
zation of a central pool of actin that associated with the MTOC and the
bulk of cellular organelles. Actin polymerization in this central region
generated pushing forces that not only deformed the surrounding
environment of the cell, but could potentially protect the nucleus and
other organelles from fatal damage[19,29,30]. This central actin pool was
controlled by the activity of the Cdc42 guanine exchange factor DOCK8
through a mechanosensitive pathway that remains to be identified.

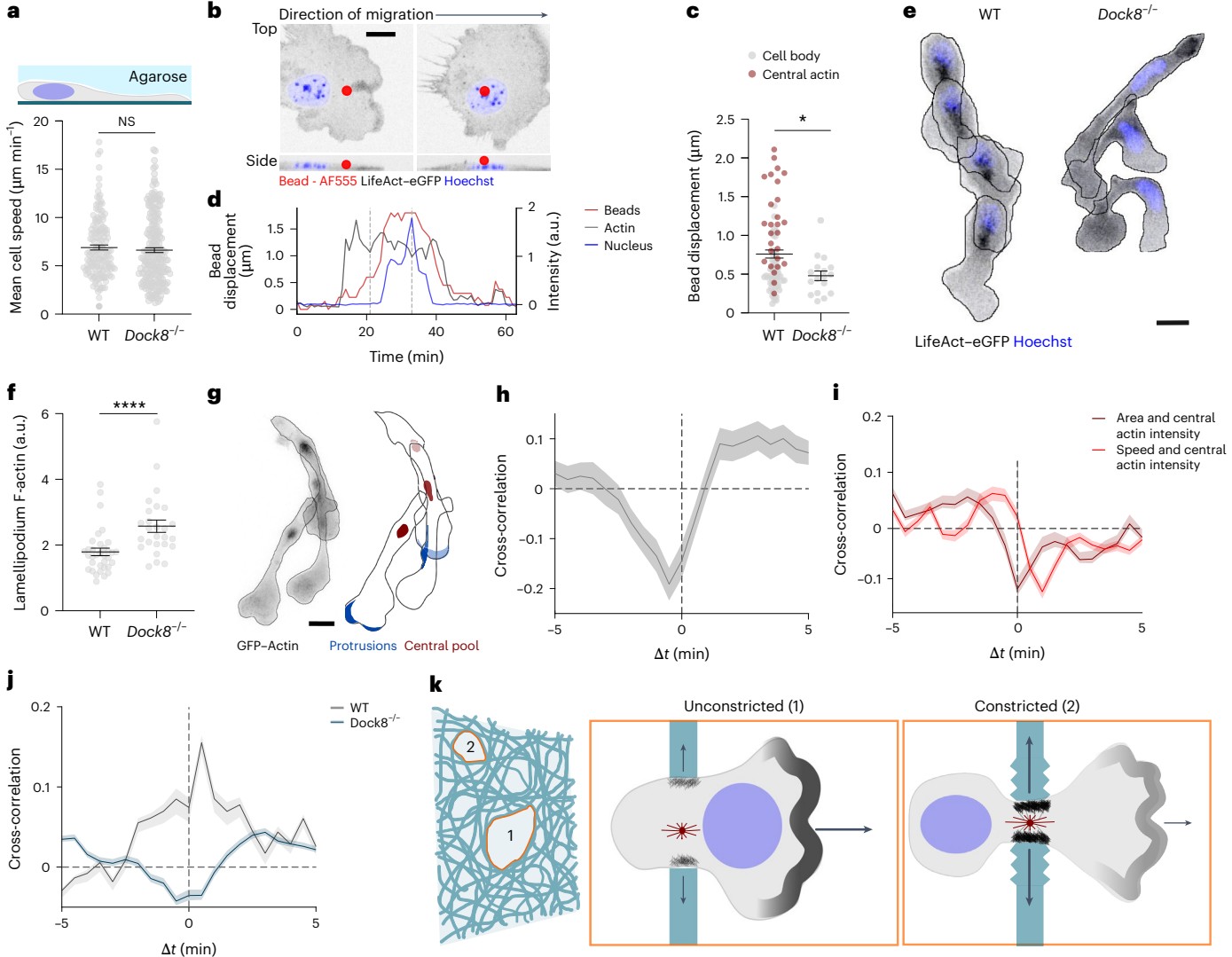

**Fig. 4 | Central actin communicates with leading edge actin. a**, Top: scheme showing DCs migrating under agarose. Bottom: mean speed of WT and *Dock8*[-/-] DCs migrating under 1.0% agarose. WT, *n* = 86 cells; *Dock8*[-/-], *n* = 111 cells from three independent experiments; *P* = 0.1658. **b**, Top view (top) and lateral projection (bottom) of *Dock8*[-/-] LifeAct–eGFP[+] (actin, black) DCs migrating under agarose with AF555[+] fluorescent beads showing cell body under the bead (left) and nucleus under the bead (right). Hoechst shows the nucleus in blue. Scale bar, 10 μm. **c**, Bead displacement generated by the cell body (excluding the nucleus) in WT or *Dock8*[-/-] DCs for beads displaced by the cell body (cell body) or beads displaced by the central pool of actin (central actin). Only cells whose nucleus also passed under the bead were included in the analysis. WT, *n* = 80 cells; *Dock8*[-/-], *n* = 17 cells from three independent experiments. **P* = 0.0239. **d**, Change in bead displacement in Z-stacks (beads) or intensity of LifeAct–eGFP (actin), and Hoechst (nucleus) in *Dock8*[-/-] LifeAct–eGFP[+] DCs over 60 min. Dashed gray lines indicate the two timepoints shown in **b**. **e**, Time-lapse projection of WT (left) and *Dock8*[-/-] (right) LifeAct–eGFP[+] (actin, black) DCs migrating under 1.0% agarose. Hoechst shows nucleus in blue. Scale bar, 15 μm. **f**, F-actin

density at the lamellipodium normalized to the total F-actin density in WT or *Dock8*[-/-] DCs migrating under 1.0% agarose. WT, *n* = 34 cells; *Dock8*[-/-], *n* = 26 cells. ****P* < 0.0001. **g**, Time-lapse projection of a GFP–actin[+] DC (left) or of segmented protrusions and central actin pool in DCs (right) migrating under 1.0% agarose. Cell contours are shown black. Scale bar, 15 μm. **h**, Temporal cross-correlation between central actin and protrusion actin intensities in GFP–actin[+] DCs (*n* = 81 cells) migrating under 1.0% agarose pooled from three independent experiments. **i**, Temporal cross-correlation between central actin intensity and cell speed or cell area in LifeAct–eGFP[+] DCs pooled from three independent experiments; *n* = 68 cells. **j**, Temporal cross-correlation between cell area and cell speed in WT and *Dock8*[-/-] LifeAct–eGFP[+] DCs migrating under 1.0% agarose pooled from three independent experiments. WT, *n* = 68 cells: *Dock8*[-/-] *n* = 177 cells. **k**, Scheme showing DCs migrating with a nucleus-first configuration and an actin enriched leading edge in unconstricted environments (1) or DCs with recruitment of actin to the central pool, which promotes deformation of the surrounding environment and reduction of actin at the leading edge in constricted environments (2). Error bars in **a**, **c**, **f**, **h**, **i** and **j** are s.e.m. Two-tailed Man–Whitney test (**a**,**c**,**f**).

An upstream activator of DOCK8 is the Hippo kinase MST1, raising the possibility that this key mechanosensitive pathway that regulates organ shape and size through controlling cell proliferation, might also control the shape of cells through its noncanonical effectors DOCK8 and Cdc42 (ref. 31).

How different actin pools communicate is poorly understood in animal cells[32], but better studied in yeast, where F-actin forms either patches or cables. In yeast, reduction of one structure is balanced by the

increase of the other, leaving the overall levels of F-actin conserved[33]. We found a similar homeostatic balance in DCs and described a regulatory loop between the central and the leading-edge pools of actin. Our results suggest that, through this communication axis, cells can coordinate protrusions in two orthogonal directions. Accordingly, DOCK8-deficient cells that lack this coordination fragment because a chemotactically enhanced leading edge loses contact with an immobilized cell body that is unable to push obstacles away. Together, our

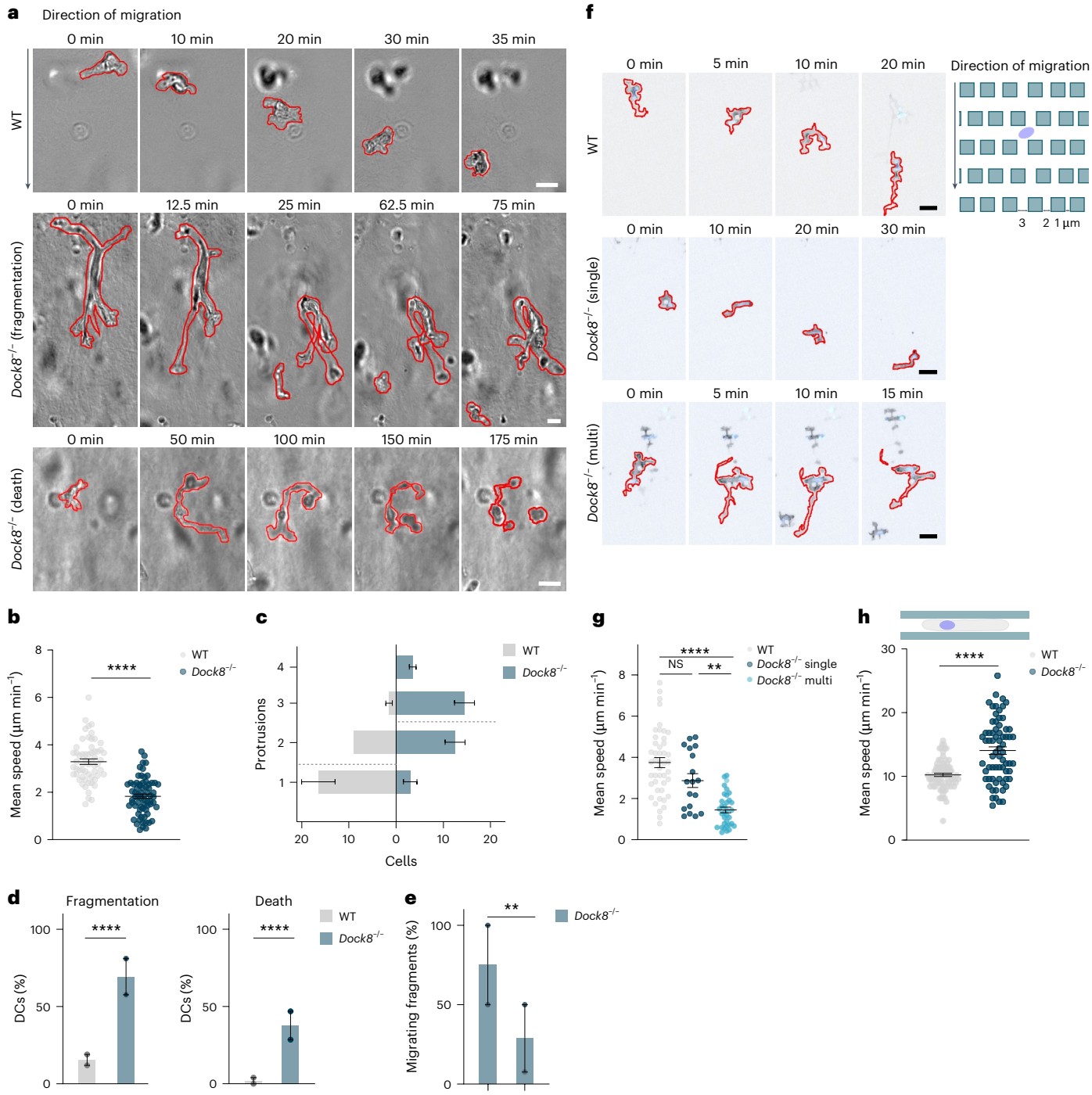

**Fig. 5 | DOCK8 affects DC locomotion depending on environmental factors.**
**a**, Representative images of WT, (top) and *Dock8*[−/−] (middle, bottom) LifeAct–eGFP[+] (actin, black) DCs migrating in 1.7 mg ml[−1] collagen showing representative cell shapes during migration (0–35 min, top) fragmentation (0–75 min, middle) or apoptosis (0–175 min, bottom). Cell contour is shown in red. Scale bar, 10 μm **b**, Mean speed of WT and *Dock8*[−/−] DCs migrating in 1.7 mg ml[−1] collagen over 4–6 h. WT, *n* = 59 cells; *Dock8*[−/−], *n* = 69 cells, from two independent experiments. ****P < 0.0001. **c**, Number of simultaneous protrusions in WT and *Dock8*[−/−] DCs migrating as in **a**. Dashed lines, overall mean number of protrusions observed during migration. WT, *n* = 54 cells; *Dock8*[−/−], *n* = 67 cells from two independent experiments; *P* < 0.0001. **d**, Fragmentation (left) and death (right) rates in WT and *Dock8*[−/−] DCs migrating as in **a**. WT, *n* = 54 cells; *Dock8*[−/−], *n* = 67 cells from two independent experiments. ****P < 0.0001. **e**, Percentage of migrating fragments originated from protrusions (Front, *n* = 11 fragments) or the rear (Rear, *n* = 25 fragments) of *Dock8*[−/−] DCs in two independent experiments. **P = 0.0042.

**f**, Representative images of WT (top) and *Dock8*[−/−] (middle, bottom) LifeAct–eGFP[+] (actin, black) DCs labeled with Hoechst (nucleus, blue) moving in pillar mazes with 6 μm in height and 1-μm, 2-μm or 3-μm-distanced pillars (top right) showing *Dock8*[−/−] DC with a monopolar configuration (*Dock8*[−/−] single, middle) and with several lamellipodia (*Dock8*[−/−] multi, bottom). Cell contour is shown in red. Scale bar, 20 μm. **g**, Migration speed of WT and *Dock8*[−/−] DCs with either single lamellipodium (*Dock8*[−/−] single) or several lamellipodia (*Dock8*[−/−] multi) migrating as in **f**. WT, *n* = 45 cells, *Dock8*[−/−] single, *n* = 18 cells, and *Dock8*[−/−] multi, *n* = 34 cells from three independent experiments. ****P < 0.0001; **P = 0.0033; NS, *P* = 0.2748. **h**, Schematic of DC (top) and mean speed of WT and *Dock8*[−/−] DCs (bottom) in straight PDMS channels. WT, *n* = 83 cells; *Dock8*[−/−], *n* = 69 cells in three independent experiments ****P < 0.0001. Error bars in **b**, **g** and **h** are s.e.m. Histogram bars in **c**, **d** and **e** are mean ± s.e.m. Two-tailed unpaired *t*-test (**b**,**h**). Two-sided Fisher's exact test (**d**,**e**). Two-tailed unpaired Mann–Whitney test (**g**).

findings establish a regulatory loop between cell front and cell body that is essential for maintaining cellular coherence.

## Online content

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

## Methods

### Mouse strains

In this study, the following mice lines were used: C57BL/6 (Janvier); LifeAct–eGFP[34]; Dock8[-/-][26]; Wasp[-/-] (B6.129S6-Wastm1Sbs/J; cat. no. 019458; The Jackson Laboratory). All mice used were bred on a C57BL/6 background and maintained at the Institute of Science and Technology Austria Institutional animal facility following the guidelines from its ethics commission and the Austrian law for animal experimentation.

### Generation and maintenance of immortalized hematopoietic progenitor cells

Hematopoietic progenitor cells were generated from the isolated bone marrow of 8- to 10-week-old mice that were retrovirally infected with an estrogen-regulated HoxB8 as described previously[17,35]. Conditionally immortalized early hematopoietic progenitor cells were kept in R10 medium (RMPI 1640 supplemented with 10% fetal calf serum (FCS), 2 mM L-glutamine, 100 U ml[-1] penicillin, 100 μg ml[-1] streptomycin and 50 μM β-mercaptoethanol (all Invitrogen), supplemented with 0.01% β-estradiol and 5% of in-house-generated Flt3l-containing supernatant). All cells were kept at 37 °C and 5% $CO_2$ until differentiation.

### Constructs used for reporter progenitor cell lines

eGFP–Centrin positive cells were generated from a human centrin1 construct (a gift from A.-M. Lennon-Dumenil's laboratory)[36]. eGFP–WASp[37], LifeAct–mCherry, LifeAct–GFP[38], EMTB–mCherry and EB3–mCherry expressing DCs[39] were generated as described[40]. GFP and DOCK8–GFP plasmids and GFP–actin plasmid[41] (a gift from M. Davidson to Addgene, plasmid no. 56421) were modified to a pLenti6.3 backbone using Gibson Assembly strategy.

### Lentivirus production and transduction into progenitor cells

Fusion-protein-coding lentiviruses were produced in Lenti-X-293 cells derived from HEK293 cells (TakaraBio). Lenti-X-293 cells were maintained in DMEM (Invitrogen) at 37 °C and 5% $CO_2$ and transfected with the above-mentioned plasmids and two helper plasmids in OptiMEM (Invitrogen) and PEI (1 mg ml[-1], Polysciences). The supernatant was collected 48 h after transfection and the resulting lentivirus preparation was concentrated using Lenti-X Concentrator (Clontech) according to the manufacturer's instructions. Progenitor cells were transduced with the concentrated lentiviral preparations by spin infection (1,500g, 1 h) with 8 μg ml[-1] Polybrene. Cells expressing the virus insertion were sorted in a Sony SH800 SFP cell sorter (sorting chip: 100 μm) for mCherry or GFP expression before DC differentiation.

### CRISPR-Cas9 ribonucleoprotein electroporation for generation Talin 1 knock-out precursor cells

Synthetic guide RNAs (crRNAs)[42] were designed using the Horizon Discovery online tool (https://horizondiscovery.com/en/ordering-and-calculation-tools/crispr-guide-rna-designer), targeting exon 25 of the mouse gene encoding Talin 1 (Tln1). crRNA sequences: Talin 1 control (Ctrl): nontargeting control 1 (Horizon Discovery); Talin 1 knock-out (Tln1[-/-]): s(5'–3') CTCACTGTTTCCCCGGGTA[18]. Precursor cells were generated following manufacture's instructions. Briefly, $1 \times 10^6$ precursor cells were collected by centrifugation, washed with PBS and resuspended in 100 μl of OptiMEM (Invitrogen). A mix of tracrRNA, crRNA and Cas9 (all Horizon Discovery) was added to the cell suspension and transferred to an electroporation cuvette. The mixture was electroporated using a specifically designed protocol (program A30) with an Amaxa nucleofector (Lonza) and transferred promptly to a well-plate prewarmed at 37 °C and 5% $CO_2$. Cells were further incubated for 72 h before being single-cell-sorted with a Sony SH800 SFP cell sorter (sorting chip: 100 μm). Single-cell clones were tested as described[40] and further confirmed by sequencing of the region of interest.

### Purification and maintenance of T cells

T cells were isolated from the spleens C57BL/6J, LifeAct–eGFP mice using an EasySep Mouse T cell Isolation Kit (STEMCELL Technologies, cat. no. 19851) according to the manufacturer's instructions. Isolated T cells were plated on cell-culture wells coated with anti-CD3e and anti-CD28 antibodies (1 μg ml[-1], Invitrogen, cat. nos. 16-0031-82, RRID:AB_468847 and 16-0281-82, RRID:AB_468921) for 2 days in R10 medium supplemented with interleukin-2 (IL-2) (10 ng ml[-1]; R&D Systems). Activated T cells were collected and expanded in IL-2-containing R10. Activated T cells were kept at 37 °C and 5% $CO_2$ for a maximum of 1 week.

### Differentiation and maturation of DCs

DCs, with the exception of Tln1[+/+] and Tln1[-/-] DCs, were differentiated by seeding $3 \times 10^5$ precursor cells in a 10 ml dish containing R10 medium supplemented with 10% of in-house-generated granulocyte-macrophage colony-stimulating factor (GM-CSF) hybridoma supernatant. On the third day of differentiation, 10 ml of R10 medium containing 20% GM-CSF was added to each dish. Half of the medium was replaced with R10 medium containing 20% GM-CSF on day 6 and cells were either harvested for maturation or frozen on day 8.

Tln1[+/+] and Tln1[-/-] DCs were differentiated by seeding $1 \times 10^5$ and $3 \times 10^5$ precursor cells, respectively, in a 10 ml dish containing R10 medium supplemented with 10% of in-house-generated GM-CSF hybridoma supernatant and 1% of in-house-generated Flt3l-containing supernatant. On the third day of differentiation, all medium was removed and 20 ml of R10 medium containing 20% GM-CSF was added to each dish. On day 6, medium was replaced as described above and cells were frozen or harvested for maturation on day 8.

Maturation of all DCs was induced by overnight stimulation with lipopolysaccharide (LPS) from Escherichia coli 0127:B8 (Sigma) at a final concentration of 200 ng ml[-1].

### Enucleation of mature DCs

Enucleation of mature DCs was performed as described previously[43]. Briefly, from a 50% (v/v) solution Ficoll-400 (Fisher Scientific), prepared with phosphate-buffered saline (PBS), a 30% (v/v) stock solution was made with D10 (DMEM supplemented with 10% FBS and 100 μl ml[-1] Pen-strep, all Invitrogen). The stock solution was filtered with a 0.4 PES filter and diluted using D10 supplemented with cytochalasin B (10 mg ml[-1]) (Tocris) and dimethylsulfoxide (DMSO) (0.2%), to final concentrations of 20%, 18% and 15%; 2 ml of each of these solutions was layered into an ultracentrifuge tube (13.2 ml thin wall, ThermoFisher Scientific) from the most to the least concentrated. The tube was covered and the gradient was incubated at 37 °C overnight. Next day, $1–2 \times 10^7$ matured DCs resuspended in 1 ml of prewarmed 15% Ficoll were added on top of the gradient. The tube was filled with D10 medium containing cytochalasin (10 mg ml[-1]) and loaded into a prewarmed (31 °C) SW641 rotor of a Sorval wx100 (Thermos Scientific). Cells were centrifuged for 1 h at 27,000 rpm (started with acceleration of 9 and stopped with deceleration of 1). After centrifugation, cells were extracted and washed three times with PBS (5 min, 300g). Cells were then resuspended in 1 ml of R10 medium, labeled with NucBlue (Life Technologies) and incubated at 37 °C and 5% $CO_2$ for at least 30 min before use.

### Flow cytometry analysis of DCs

DCs were checked routinely for correct surface expression markers using antibodies against MHCII and CD11c (cat. nos. 48-5321-82 and 17-0114-82, respectively, both eBiosciences). Stainings were performed in fluorescence-activated cell sorting (FACS) buffer (1× PBS, 2 mM EDTA, 1% BSA) with Fc receptor blockage (anti-mouse CD16/CD32, BioLegend). Analysis was carried either on a FACSCanto BD Biosciences or in a BC CytoFLEX LX.

## Transient transfection of DCs

The following plasmids were used: eGFP, pcDNA3-EGFPCdc42(WT), and pcDNA3-EGFP-Cdc42(T17N)[44] (gifts from K. Hahn to Addgene, plasmids nos. 12599 and 12601). DCs derived from progenitor cells were transfected with 4 µg of DNA using the nucleofector kit for primary T cells (Amaxa, Lonza Group) following the manufacturer's guidelines. Briefly, $4-5 \times 10^6$ cells were resuspended in 100 µl of DMEM (Invitrogen) and 4 µg of plasmid DNA. Cells were transferred to a cuvette and electroporated using a specifically designed protocol (program X-001). Transfected DCs were incubated overnight in R10 supplemented with 10% GM-CSF and LPS (200 ng ml$^{-1}$). Experiments were carried out the next day, and only GFP-expressing cells were analyzed.

## Pharmacological inhibitors

The following small molecule inhibitors were used: ZCL278 (ref. 45) (MedChemExpress) and ML141 (ref. 46) (Sigma) to perturb Cdc42 activity; and NSC23766 (MedChemExpress) to perturb Rac1 activity[47]. Inhibitors were diluted in DMSO, mixed with the DC suspension after maturation for at least 30 min and kept through the assays at the final concentration indicated (ZCL278, 10 µM; ML141, 20 µM; NSC23766, 50 µM).

## FACS F-actin analysis in DCs

After overnight stimulation with LPS, WT and *Dock8*$^{-/-}$ DCs were recovered in 12-well plates in 500 µl R10 for 30 min at 37 °C. Cells were stained during fixation (4% paraformaldehyde (PFA), 20 µM FITC-phalloidin and 0.5% saponin in PBS, 500 µl, 20 min at 37 °C) and analyzed on a FACS Aria III. Stainings were carried out in three biologically independent samples.

## Immunodetection of whole-cell lysates

DCs ($1.6 \times 10^5$) were harvested and washed with PBS. The cell pellet was lysed using RIPA buffer (Cell Signaling), mixed in a 1:1 proportion with 2x Laemmli buffer (Sigma) and incubated for 5 min at 90 °C. Boiled samples were loaded on precast 3%–8% Tris-Acetate gel (Invitrogen) and ran in 1× Tris-acetate running buffer. Resulting samples was transferred to a nitrocellulose membrane using the iBlot Dry Blotting System (Invitrogen) and blocked for 1 h with 5% powder milk in TBS containing 0.01% Tween-20. For whole-cell lysate protein detection, the following primary antibodies were used: mouse monoclonal anti-talin antibody (1:400 dilution, cat. no. T3287, Sigma), and rabbit polyclonal HSPA1A (anti-HSP70) antibody (1:10,000 dilution, cat. no. PA5-34772, ThermoFisher Scientific). Membranes were incubated with the primary antibody solutions overnight at 4 °C. Membranes were then washed and incubated with the secondary antibody solutions for 1 h at room temperature. The following secondary antibodies were used: goat anti-mouse IgG (H/L):HRP (1:10,000 dilution, Bio-Rad), and goat anti-rabbit IgG (H/L):HRP (1:3,000 dilution, Bio-Rad). Enzymatic reaction was started by addition of chemoluminescent substrate for HRP using Clarity Western ECL substrate and acquired on a ChemieDoc MP imaging system (all Bio-Rad).

## Adhesion assay

WT, *Tln1*$^{+/+}$ and *Tln1*$^{-/-}$ DCs were matured with LPS as described above. Upon addition of LPS, cells were seeded in a TPP 10 cm$^2$ round tissue culture plate (Sigma) and incubated at 37 °C and 5% CO$_2$ for 45 min. Low magnification images were acquired in an inverted DM IL Led Fluo Leica Microsystems microscope, using a ×20/0.3 numerical aperture (NA) air objective equipped with an iDS U3-36PxXCP-C camera. Cells were considered adherent based on their morphology.

## Under-agarose migration assay of mature DCs

Glass coverslips were glued to the bottom of a petri dish with a 17-mm-diameter hole where a custom-made plastic ring was attached using paraffin (Paraplast X-tra; Sigma). Agarose solution was prepared by mixing one part of 2x Hank's buffered salt solution (Sigma), pH 7.3 with two parts RPMI (Invitrogen) supplemented with 20% FCS (Invitrogen) and two times the concentration of all the other supplements used

in R10 medium (see above) and either 2% or 4% of UltraPure Agarose (Invitrogen) dissolved in one part water to achieve different agarose stiffnesses. The liquid agarose (400 µl) was poured into the dish, covering the coverslip. The agarose was allowed to solidify at room temperature for 5 min, after which two holes (1.5 mm and 2.0 mm) were punched into the agarose. The dishes were incubated at 37 °C and 5% CO$_2$ for 30 min for equilibration. 2.5 µg ml$^{-1}$ of CCL19 (PrepoTech) diluted in R10 was placed in the 2 mm hole and $0.5-1 \times 10^6$ mature DCs were placed in the 1.5 mm hole opposite the chemokine. Before acquisition, dishes were incubated for at least 1 h at 37 °C and 5% CO$_2$ to allow invasion under the agarose. All images were acquired under physiological conditions using custom-built climate chambers (37 °C, 5% CO$_2$, humidified).

## Passivation of coverslips using PLL-PEG

Glass coverslips (22 × 22 mm, Fisher Scientific) were sonicated in a solution of 70% ethanol and for at least 15 min. After sonication, coverslips were air-dried glued to the bottom of a petri dish with a 17 mm diameter hole where a custom-made plastic ring was attached using paraffin (Paraplast X-tra; Sigma). Coverslips were then covered with a solution of PLL-PEG (1 mg ml$^{-1}$, SuSoS Surface Technology) overnight at 4 °C. After incubation, coverslips were washed at least three times with PBS. Dishes were further assembled for the under-agarose assay as described above.

## Immunofluorescence under agarose

For analysis of fixed samples, a round-shaped coverslip (cat. no. 1.5, 10 mm, Mentzel, ThermoFisher Scientific) was placed in a glass-bottom dish before casting the agarose. DCs and chemokine were added to the dishes as described above. Cells were allowed to invade and migrate for at least 3 h at 37 °C and 5% CO$_2$. Migrating cells were fixed with prewarmed PBS supplemented with 4% PFA for 20 min at 37 °C. After fixation, the agarose patch was removed carefully and the coverslip recovered and thoroughly washed with PBS. Coverslips were incubated with 0.1% Triton X-100 in PBS for 20 min at room temperature, washed with PBS, and blocked with 1% bovine serum albumin (BSA) for 1 h at room temperature. Primary antibodies were diluted in PBS with 1% BSA and incubated either overnight at 4 °C or for 2 h at room temperature. After primary antibody incubation, cells were washed three times with PBS and incubated with secondary antibody diluted in PBS with 1% BSA for 1 h at room temperature. Stained coverslips were washed three times with PBS and mounted on a slide using Flourmount-G mounting medium with DAPI (cat. no. 00-4959-52, ThermoFisher Scientific). Slides were imaged the next day or stored at 4 °C in the dark until image acquisition.

Confocal imaging of fixed samples was performed using an upright confocal microscope plus Airyscan (cat. no. LSM800, Zeiss) equipped with two GaAsP Photomultiplier detectors using a ×40/1.3 oil differential interference contrast (DIC), ultraviolet–infrared objective. Multipositions of Z-stacks (0.4-µm step size) of fixed migrating cells were acquired using Zeiss software (ZEN v.3.8).

## Primary and secondary antibodies

The primary and secondary antibodies used are as follows:

| Target | Primary antibodies | Secondary antibodies |
|---|---|---|
| MTOC | anti-γTubulin (abcam, cat. no. ab11317, 1:400) | Goat anti-Rabbit IgG (H+L) Alexa Fluor$^T$ 488 (Invitrogen, cat. no. A-11008, 1:200) |
| Golgi complex | anti-Giantin (Sysy antibodies, cat. no. 263003, 1:100) | |
| Lysosomes | anti-LAMP2 (abcam, cat. no. ab13524, 1:100) | Donkey anti-Rat IgG (H+L) Alexa Fluor 488 (Jackson Immuno Research, cat. no. AB_2340686, 1:200) |
| F-actin | Alexa Flour 647 Phalloidin (Invitrogen, cat. no. A22287, 1:400) | |

## Two-dimensional analysis of cells migrating under agarose

For two-dimensional analysis of cells migrating under agarose, LifeAct–eGFP or actin–eGFP expressing DCs labeled with Hoechst (NucBlue, cat. no. Hoechst 33342, Invitrogen) were imaged with an inverted widefield Nikon TiE2 microscope equipped with either a ×20/0.75 DIC 1 air PFS or a ×40/0.95 NA DIC air PFS objectives using a DS-Qi2 CMOS monochrome camera and a Lumencor Spectra III light source (390/22 nm, 440/20 nm, 475/28 nm, 511/16 nm, 555/28 nm, 575/25 nm, 635/22 nm, 747/11 nm; Lumencor). Images were taken every 30 s at multipositions using the NIS Elements software (Nikon Instruments).

Single cells moving under agarose and their central actin pool were segmented based on either the LifeAct–eGFP or the GFP–actin signal using Ilastik pixel classification[48] and tracked using Fiji-Trackmate[49]. Resulting tracks were curated manually and only noninteracting, well-isolated cells with tracks longer than ten frames (5 min) were processed further. Cell speed was calculated for LifeAct–eGFP cells using their center of mass, based on the outline generated by the segmentation. Protrusion areas were defined by the GFP–actin signal and correspond to the nonoverlapping regions of the cell segmentations at times $t$ and $t+1$. All actin intensities are the integrated and background-corrected to the actin signal in the respective areas. Comparison of the temporal cross-correlation of two parameters (central actin intensity, cell area, cell speed) and test for the statistical significance of the temporal offset we used cross-correlation analysis[50] with a custom-written MATLAB script (MATLAB, R2020a).

## Total internal reflection microscopy of cells migrating under agarose

Total internal reflection microscopy (TIRF) imaging of DCs migrating under agarose was performed at 37 °C and 5% $CO_2$ using a Zeiss Axio Observer.Z1 inverted fluorescence microscope equipped with a ×63/1.46 oil TIRF (WD 0.10 mm) objective, four fiber-coupled laser for TIRF (405 nm, 488 nm, 561 nm and 640 nm), and ×2 photometric Evolve 512 EM-CCD cameras. Images were acquired every second for at least 10 min using VisiView software (Visitron). Resulting movies were further processed using Fiji/ImageJ.

## Bead displacement in agarose

To track the force cells exert on the agarose, polystyrene microspheres with a nominal diameter of 1 µm and labeled with a fluorescent red dye (Red-580/605, cat. no. F-13083, Invitrogen) were added to the agarose solution (1:100 dilution). The agarose cast and cell addition were performed as mentioned above.

Imaging of LifeAct–eGFP expressing DCs labeled with Hoechst (NucBlue, cat. no. Hoechst 33342, Invitrogen) was performed under physiological conditions using custom-built climate chambers (37 °C, 5% $CO_2$, humidified) on an inverted spinning-disc confocal microscope (Nikon CSU-W1) cameras using a ×40/1.15 water objective. Z-stacks (0.1 µm step size) of migrating cells were acquired using two teledyne photometric BSI (USB3) sCMOS cameras with 95% quantum efficiency and a 6.5 × 6.5-µm pixel area. Images were acquired every 30 s for approximately 20 min.

Bead displacement analysis was performed using custom Python scripts. First, beads were segmented individually and labeled using their maximum intensity projection in $Z$ and time to discard nonstationary beads, and a size filter was used to exclude bead aggregates. Next, we defined a fixed volume around each bead spanning the whole $Z$-stack and five by five pixels in $XY$. To track the movement in $Z$, we generated a time kymograph of the bead intensity projected along the $x$ and $y$ axes and tracked the moving bead edge, detected using the Otsu threshold method (Extended Data Fig. 2a). We then computed the total actin and nuclear intensity in each time frame within a similar volume of 20 × 20 pixels in XY, centered around each bead. To classify the actin contribution between none, cytoplasmic and central actin, we ran a K-means clustering algorithm with two ($Dock8^{-/-}$ DCs) or three (WT DCs) clusters on the actin intensity curves. Hoechst (blue) signal

was used to classify the nuclear contribution to the displacement. We then computed the average position of the bead for each of the regions and subtracted it from the baseline (no cell).

## Manufacturing and migration assay in polydimethylsiloxane height confiners

The microfabricated polydimethylsiloxane (PMDS) devices used to confined the cells in environments with different heights consist of two glass coverslips spaced by PDMS micropillars. One of the glass coverslips (no. 1.5, 22 × 22 mm, Mentzel, ThermoFisher Scientific) was glued to a petri dish with a 17-mm-diameter hole using aquarium sealant, and the other, containing the PDMS micropillars, was attached to a PDMS cylinder secured by a magnetic device.

The pattern mold was produced by photolithography on a silicon wafer. The wafer was coated with SU8-GM1050 (Gersteltec) and soft-baked for 1 min at 120 °C, followed by 5 min at 95 °C. The wafer was developed in SU8 developer for 17 s and then silanized with trichloro(1H,1H,2H,2H-perfluorooctyl)silane in a vacuum desiccator for 1 h.

To produce the PDMS piston, silicone elastomer and curing agent were mixed in a 30:1 ratio, degassed as described before, and poured into an aluminum mold with the required dimensions. The PDMS pistons were cured at 80 °C for 6 h and peeled off the silicon wafer with isopropanol.

Micropillars were produced by mixing silicone elastomer and curing reagent (PDMS Sylgard 184 Elastomere Kit, Dow Corning) in a 7:1 ratio. The mixture was then degassed using a planetary centrifugal mixer (ARE250, Thinky) and poured carefully onto the wafer. Round coverslips (no. 1.5, 10-mm diameter, Mentzel, ThermoFisher Scientific) were plasma activated for 2 min (Plasma Cleaner, Harrick Plasma) and placed on the wafer with the activated surface facing the elastomere/curing agent mixture. The wafer was cured on a heating plate for 15 min at 95 °C, and the micropillar-coated coverslips were removed with a sharp razor blade and isopropanol.

The confiner devices were assembled by mounting a micropillars-bearing coverslip onto the PDMS piston with the micropillars facing upward and stuck to the glass bottom of the magnetic device. R10 medium was added to the micropillars-bearing coverslip and the petri dish containing the second glass coverslip, and incubated at 37 °C and 5% $CO_2$ for at least 30 min to equilibrate. Matured DCs were resuspended in R10 with 2.5 µg ml$^{-1}$ of CCL19 (PrepoTech) in a final volume of 20 µl and added to the micropillars. The PDMS piston with the micropillars and the cell mixture were then pressed onto the glass coverslip in the petri dish and sealed by a metal ring. Confined cells were incubated for at least 1 h at 37 °C and 5% $CO_2$ before imaging.

Imaging was performed as described in 'bead displacement under agarose.' Z-stacks (0.4-µm step size) of migrating cells were every 30 s for approximately 20 min.

## Manufacturing and migration assays in PDMS pillar mazes, straight and constricted channels

The microfabricated PDMS devices containing pillar forest, straight or constricted channels consist of PDMS blocks (fabricated as above, but using a 1:10 elastomer to curing agent ratio) attached to one glass coverslip. The devices were then cut in small squares (approximately 1 × 1 cm$^2$) and attached to plasma-cleaned coverslips (no. 1.5, 22 × 22 mm, ThermoFisher Scientific), and incubated at 85 °C for 1 h.

The coverslips with the PDMS devices were then glued to a petri dish containing a 17-mm-diameter hole using aquarium sealant. Before adding the cells, devices were flushed and incubated with R10 medium for at least 1 h at 37 °C and 5% $CO_2$. Matured DCs 0.5–1 × 10$^6$ were added to one side of the devices and R10 with 2.5 µg ml$^{-1}$ of CCL19 (PrepoTech) was added to the opposite side. Cells were incubated for at least 1 h before image acquisition.

## Analysis of the central actin pool intensity changes in cells moving in pillar mazes

Widefield images of LifeAct–eGFP expressing DCs moving in pillar mazes were acquired as described in '2D analysis of cells migrating under agarose.' Images were taken every 30 s at multipositions with NIS Elements software (Nikon Instruments).

Cell area and nucleus were segmented and tracked employing the Ilastik pixel classification/cell tracking workflows. Noninteracting, well-isolated cells were identified and stabilized to their center of mass. For each cell, all regions of interest were annotated manually. The F-actin intensity in these regions was normalized to the overall F-actin intensity. All retraction events were pooled by shifting the events relative to each other such that $t = 0$ marks the beginning of the retraction event and by setting the intensity in all regions to 1.

## Analysis of cells moving in straight and constricted channels

Imaging of EB3–mCherry and LifeAct–eGFP expressing DCs in straight and constricted channels was performed in an inverted widefield Nikon TiE2 microscope equipped with a ×40/0.95 NA DIC air objective using a Nikon DS-Qi2 CMOS monochrome camera and a Lumencor Spectra III light source (390/22 nm, 440/20 nm, 475/28 nm, 511/16 nm, 555/28 nm, 575/25 nm, 635/22 nm, 747/11 nm; Lumencor). Images were taken every 60 s at multipositions with NIS Elements software (Nikon Instruments).

Actin distribution in cells during constriction passage was quantified using custom scripts in Python. First, channels were segmented using the brightfield images, and the actin signal was averaged vertically ($y$ axis, only in segmented areas) to create a longitudinal actin density profile for each time frame. A maximum projection of these profiles resulted in a final time-averaged actin density profile, which was used to compute the ratio between the actin signal inside and outside the constriction.

## Collagen migration assay of mature DCs

Custom-made migration chambers were assembled using a petri dish with a 17-mm-diameter hole in the middle that was covered by two glass coverslips[51] (no. 1.5, 22 × 22 mm, ThermoFisher Scientific).

The collagen mixture, consisting of either 1.5 or 3 mg ml$^{-1}$ bovine collagen I (PureCol, Nutragen; both AdvancedBioMAtrix), was reconstituted by mixing $1.5–3.0 × 10^5$ matured DCs in suspension (R10 medium) with collagen I solution buffered to physiological pH with Minimum Essential Medium (Sigma) and sodium bicarbonate (Sigma) in a 1:2 ratio. In the experiments where labeled collagen was used, a mix of unlabeled and labeled collagen was used at a ratio of 1:2.

The collagen and cell mixture were then added to the migration chamber and allowed to polymerize in a vertical position for 1 h at 37 °C, 5% CO$_2$. Directional cell migration was induced by overlaying the polymerized gels with 0.63 µg ml$^{-1}$ CCL19 in R10. To prevent drying out of the gels, chambers were sealed with paraffin (Paraplast X-tra, Sigma).

Brightfield movies were acquired in inverted cell-culture microscopes (DM IL Led, Leica Microsystems) using either a ×10/NA or a ×40/NA air objective equipped with cameras (ECO415MVGE, SVS-Vistek) and custom-built climate chambers (37 °C, 5% CO$_2$, humidified). Images were acquired with a time interval of either 30 s or 60 s and global $y$ displacement was analyzed by a custom-made tracking tool.

## Collagen fiber displacement and F-actin accumulation analysis

To visualize collagen fibers, collagen was conjugated directly to Alexa Fluor 594 NHS Ester (Succinimidyl Ester, ThermoFisher Scientific). Collagen was added to SnakeSkin Dialysis Tubes, 10K MWCO, 16 mm (ThermoFisher Scientific), and immersed in 100 mM NaHCO$_3$ overnight at 4 °C to allow polymerization. Alexa Fluor 594 NHS Ester (1.5 mg ml$^{-1}$) was added to the polymerized collagen and incubated for 3 h. To remove the unconjugated dye, the collagen mixture was placed in 0.2% acetic acid in deionized water for further dialysis overnight at 4 °C. Labeled collagen was kept at 4 °C until use.

Movies of LifeAct–eGFP expressing DCs in Alexa-594-labeled collagen matrices were acquired on an inverted spinning-disc confocal microscope (Andor Dragonfly 505) using a ×60/1.4 NA objective and 488/561-nm laser lines in a custom-built climate chamber (37 °C under 5% CO$_2$). Z-stacks (1.5-µm step size) of migrating cells were recorded using an Andor Zyla camera (4.2 Megapixel sCMOS) every 60 s for 20–25 min.

Collagen fiber displacement was calculated using the software Davis v.8 (Lavision) applying Particle Image Velocimetry as described[22]. Computation of the closest distance between a collagen fiber deformation maxima and an F-actin intensity maxima across several time-lapse images and Z-slices was performed using a standardized Python function. Briefly, this function independently finds the local maxima of collagen fiber deformation or F-actin intensity within a predefined neighborhood radius and calculates the minimum distances between these two structures per Z-slice and timepoint.

## Fixation and immunofluorescence of collagen matrices

To visualize the nucleus-MTOC orientation during migration in different collagen matrices Centrin–eGFP expressing DCs labeled with Hoechst (NucBlue, cat. no. Hoechst 33342, Invitrogen) were seeded in the collagen mixture and collagen gels were cast as described above. At 3 h after the introduction of the CCL19 gradient, the collagen gels were isolated and bathed immediately in a PBS solution with 4% PFA for 10 min at room temperature. The fixed collagen gels were washed with PBS at least three times and incubated with Phalloidin-Atto647N (1:400 dilution, Sigma) diluted in PBS supplemented with 0.2% BSA and 0.05% saponin for 2 h at room temperature. After three more washes with PBS, the gels were mounted using Flourmount-G mounting medium with DAPI (cat. no. 00-4959-52, ThermoFisher Scientific).

Imaging was performed in an inverted confocal microscope (LSM800 inverted, Zeiss) equipped with two GaAsP photomultiplier tube detectors using a ×40/1.2 water objective. Multipositions of Z-stacks (0.5-µm step size) of fixed cells migrating in the collagen matrices were acquired using Zeiss software (ZEN v.3.8).

## Statistics and reproducibility

Statistical details for each experiment can be found in the figure legends. Appropriate controls were performed for each biological replicate. All replicates were validated independently and pooled only when all showed the same trend. Statistical analysis was conducted in Prism v.10.2.2 (GraphPad). Data was tested for normal distribution using the D'Agostino Pearson Omnibus k2 test. Normally distributed data was tested using a Student's $t$-test or ANOVA. Non-normally distributed data was tested using the Mann–Whitney test. Categorical data (for example presence/absence of the central actin pool) was tested using Fisher's exact test.

## Reporting summary

Further information on research design is available in the Nature Portfolio Reporting Summary linked to this article.

## Data availability

Datasets generated during this study are available from the corresponding author on request.

## Code availability

Script used to quantify proximity between collagen fibers displacement and actin bursts can be found at https://github.com/mirifaj/actin_deformation_cell_migration. Scripts used to quantify bead displacement induced by DCs migrating under agarose can be found at https://github.com/avesar/protrusion-forces. Scripts used for the correlative analysis between central actin pool and actin at the protrusions,

cell area and cell speed can be found at https://git.ista.ac.at/rhauschild/movingactincellseg. All custom-made scripts are available from the authors on request.

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

## Acknowledgements

This research was supported by the Scientific Service Units of ISTA through resources provided by the Imaging and Optics, Preclinical and Lab Support Facilities. In particular, we thank M. A. Symth and F. G. G. Leite, from the Virus Service Team, who helped generating the lentiviral particles used in this study. We thank all the members of the Sixt group for valuable discussions and feedback, in particular, I. Mayer, for helping with T cell isolation and Z. (P.) Li for providing the Actin–GFP DC line. We are also thankful to J. Mandl and C. Shen for their feedback during the writing of this manuscript. This work was supported by a European Research Council grant ERC-SyG 101071793 to M.S. M.J.A. was supported by an HFSP Postdoctoral Fellowship LTF 177 2021 and A.J.G. by a Lise Meitner Fellowship of the FWF (Austrian Science Fund). Y.F. was supported by the AMED-CREST (JP19gm1310005), the Medical Research Center Initiative for High Depth Omics and CURE:JPMXP1323015486 for MIB, Kyushu University.

## Author contributions

P.R.-R., A.J.G., K.V., N.C. and M.S. conceived the experiments. P.R.-R. performed and analyzed experiments with the help of N.C. and M.J.A. N.C. analyzed MTOC-nucleus position in microfabricated channels and under agarose; P.R.-R. and M.J.A. performed and analyzed bead displacement experiments in agarose. F.G. provided data of DCs migrating under different agarose stiffness. M.R. wrote image analysis scripts for quantification of the collagen I fiber deformation and actin bursts proximity. I.d.V. and P.R.-R. performed experiments in enucleated DCs. J.M. generated microfabricated channels and pillar arrays. R.H. wrote image analysis scripts for the analysis of the central actin pool in cells moving in pillar mazes, and cross-correlation analysis of central actin pool, protrusions and cell area/cell speed. Y.F. provided reagents, technical support and advice. P.R.-R., M.J.A. and M.S. wrote the manuscript, which was critically reviewed by all other authors.

## Funding

## Competing interests

The authors declare no competing interests.

## Additional information

**Extended data** is available for this paper at https://doi.org/10.1038/s41590-025-02211-w.

**Correspondence and requests for materials** should be addressed to Michael Sixt.

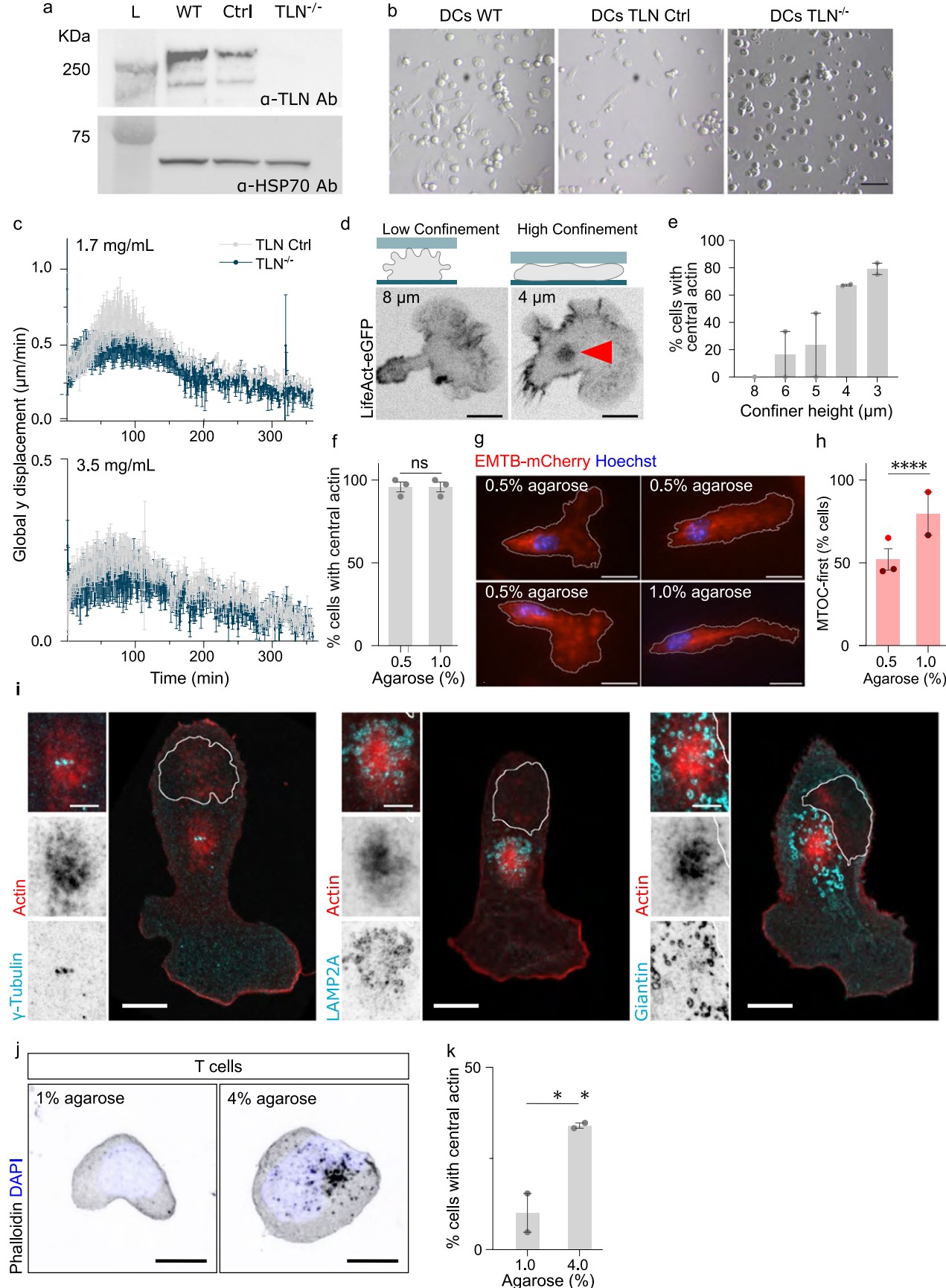

Extended Data Fig. 1 | See next page for caption.

**Extended Data Fig. 1 | Extended Data Fig. 1 related to Fig. 1. a.** Talin 1 and HSP70 detection by Western Blot in wild-type, *Tln1*$^{+/+}$ and *Tln1*$^{-/-}$ DCs. **b.** Wild-type, *Tln1*$^{+/+}$ and *Tln1*$^{-/-}$ DCs in culture dishes 45 min after treatment with lypopolysaccharide (LPS). **c.** Dots show mean global y-displacement *Tln1*$^{+/+}$ (grey) and *Tln1*$^{-/-}$ (blue) DCs migrating in 1.7 mg/mL (top) and 3.5 mg/mL (bottom) collagen gels, in three independent experiments. **d.** LifeAct-eGFP$^+$ (actin, black) DCs migrating in 8 μm (left) and 4 μm (right) vertical PDMS confiners. Actin accumulation in higher confinement is shown by the red arrow. **e.** Dots show percentage of LifeAct-eGFP$^+$ DCs showing the central actin pool in vertical confiners with heights from 8 μm to 3 μm. 8 μm, n = 10; 6 μm, n = 63; 5 μm, n = 17; 4 μm, n = 67; 3 μm, n = 14. **f.** Dots show percentage of Wild-type DCs showing a central actin pool during migration under 0.5% or 1.0% in three independent experiments. 0.5% agarose, n = 145; 1.0% agarose, n = 88. P = 0.4306. **g.** EMTB-mCherry+ (MTOC, red) DCs labeled with Hoechst (nucleus, blue) migrating under 0.5% and 1% agarose.

**h.** Dots show percentage of EMTB-mCherry$^+$ (dark red) or eGFP-Centrin$^+$ (red) DCs showing MTOC-first orientation when migrating under 0.5% or 1% agarose in at least two independent experiments. 0.5% agarose, n = 200; 1% agarose, n = 74. **** P < 0.0001. **h.** Wild-type DCs migrating under 1.0% agarose and stained with Phalloidin (F-actin, red) and antibodies specific for γ-Tubulin (MTOC - cyan, left), LAMP2 (Lysosomes – cyan, middle) or Giantin (Golgi complex - cyan, right). Nucleus position is shown by the white overlay. **i.** Primary T cells migrating under 1.0% and 4.0% agarose fixed and stained with phalloidin (F-actin, black) and DAPI (nucleus, blue). **j.** Dots show percentage of primary T cells with a central pool of actin under 1.0% and 4.0% agarose in two independent experiments. 1.0% agarose, n = 34; 4.0% agarose, n = 69. ** P = 0.0044. Histogram bars (**e**, **f**, **h**, **k**) are mean ± s.e.m. Error bars in (**c**) are s.e.m. Two-sided Fisher's exact test is used in (**f**, **h**, **k**). Scale bars in (**d**, **h**, **i**) are 10 μm, 3 μm in the inset in (**h**), and 20 μm in (**g**). n indicates number of cells.

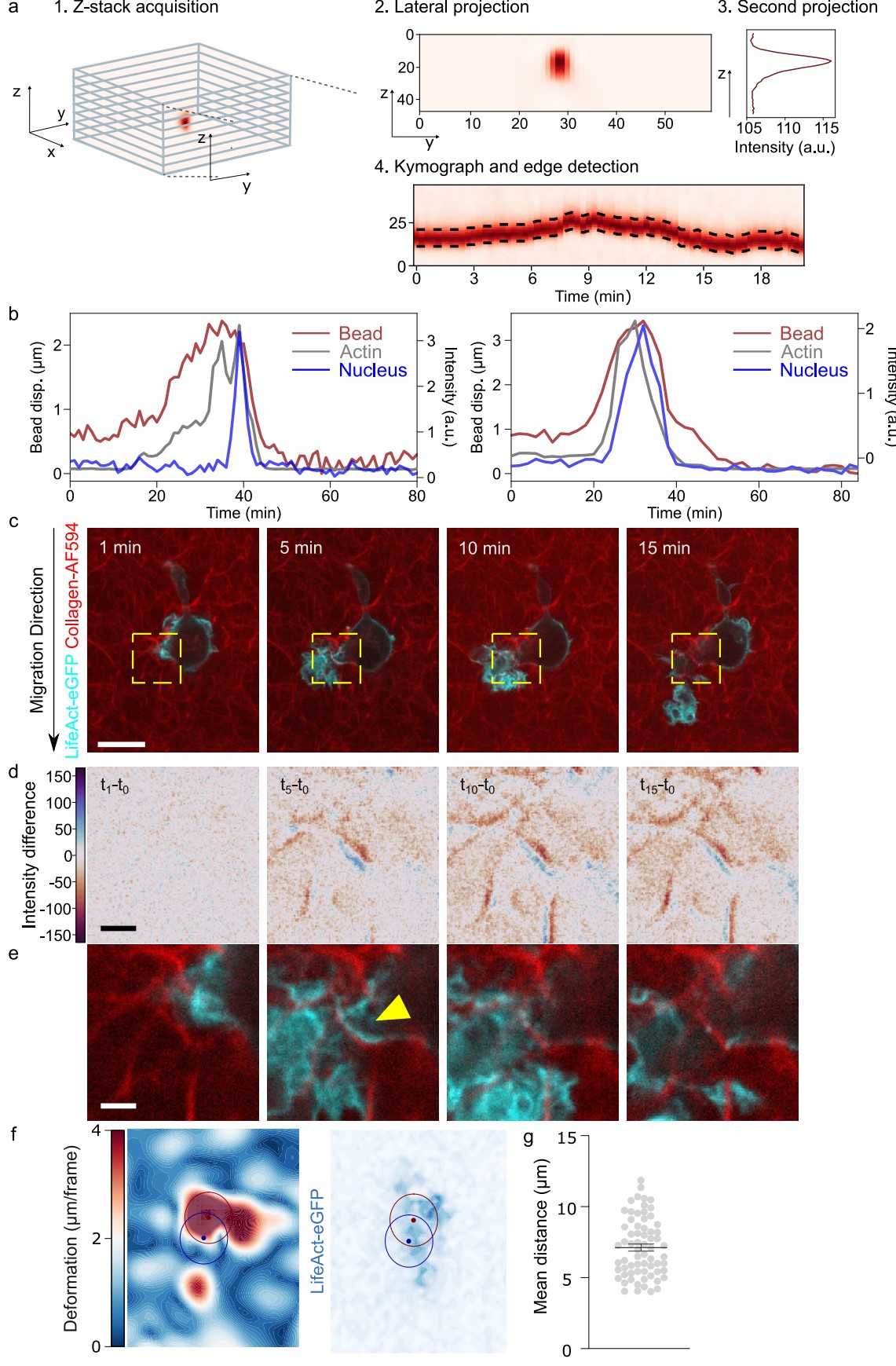

**Extended Data Fig. 2 | See next page for caption.**

**Extended Data Fig. 2 | Extended Data Fig. 2 related to Fig. 2. a.** Workflow implemented for the quantification of bead displacement. (1) Z-stacks acquisition of fluorescent beads embedded in agarose; (2) lateral projection of the resultant Z-stack; (3) intensity projection of the generated lateral projection; and (4) kymograph and bead edge detection. **b.** Change in bead displacement in Z (red) or LifeAct-eGFP (actin, grey) and Hoechst (nucleus, blue) intensities over time for two different cells. **c.** One Z-plane of a LifeAct-eGFP⁺ (actin, cyan) DCs migrating in 3.5 mg/mL collagen matrices labelled with AlexaFlour594 (red). Yellow dashed box shows the area for the insets in (**c**) and (**d**). **d.** Image subtraction of different time-points of the collagen fibers in the area highlighted by the yellow box in (**b**) shows fiber displacement over time. Red: fiber position in the initial time point; blue: same fiber in the final time point; white: no changes in fiber position. **e.** Collagen fibers and actin in the area highlighted by the yellow box in (**b**) as in (**b**). Yellow arrow shows actin accumulation at the place of collagen fiber deformation. **f.** Example of the workflow used for the quantification of the distance between collagen fiber displacement and actin accumulation. Left: Collagen fiber deformation measured by Particle Image Velocimetry (PIV). Dark red shows high deformation while dark blue no deformation areas. Right: LifeAct-eGFP intensity. Dark red dot shows the maximum deformation observed in collagen fibers and dark blue dot the maximum LifeaAct-eGFP intensity. **g.** Dots show mean distances between maximum collagen fiber deformation and maximum actin spots as in (**e**) in three independent experiments. n = 10 cells. Error bars show s.e.m. Scale bars in (**d**, **e**) are 2 μm and 10 μm in (**c**).

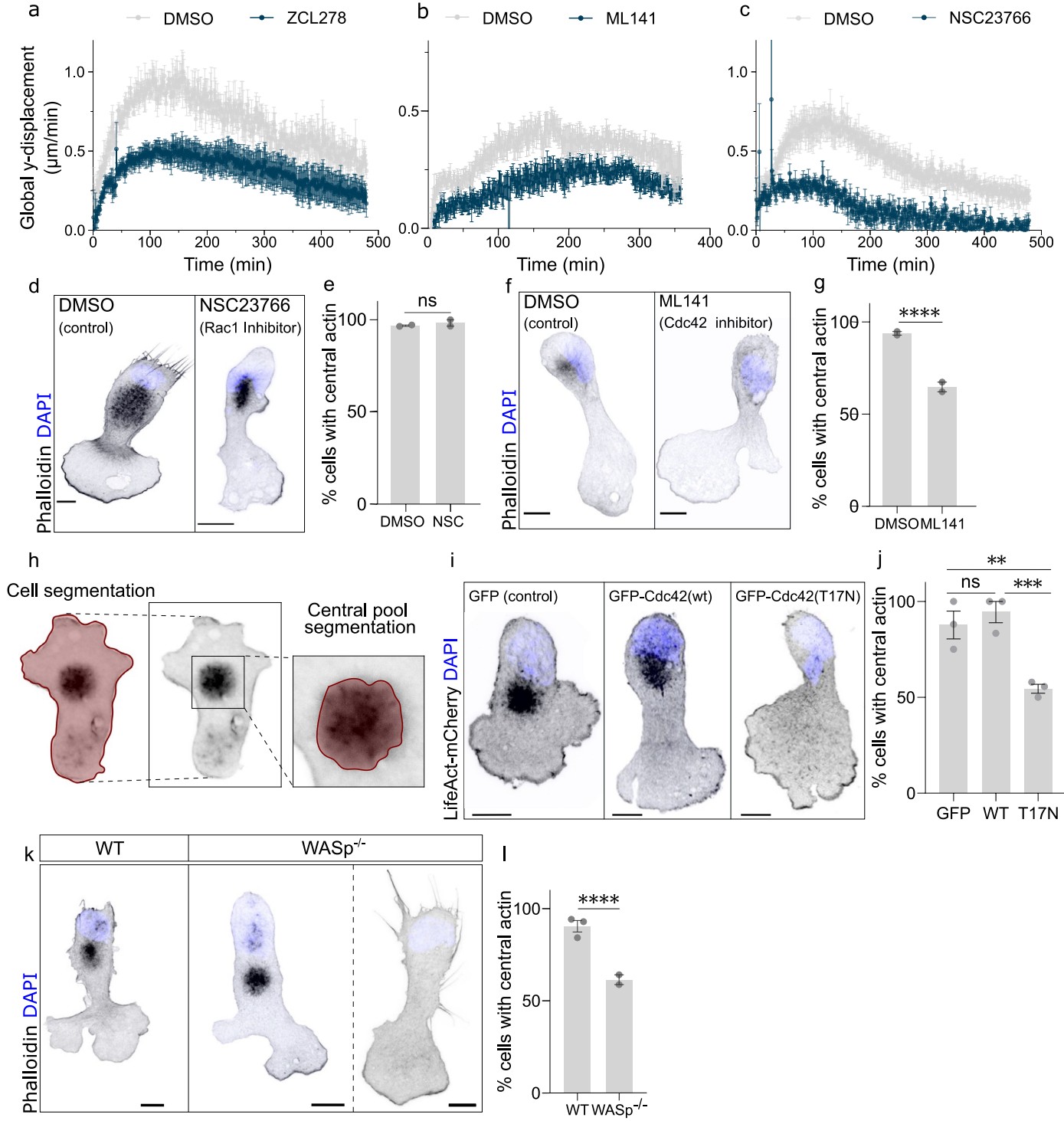

**Extended Data Fig. 3 | Extended Data Fig. 3 related to Fig. 3. a.** Dots show mean global y-displacement (speed) of wild-type DCs treated with DMSO (gray) or ZCL278 (blue) migrating in 1.7 mg/mL collagen gels in three independent experiments. n = 6. **b.** Dots show mean global y-displacement (speed) of wild-type DCs treated with DMSO (gray) or ML141 (blue) as in (**a**). n = 6. **c.** Dots show mean global y-displacement (speed) of wild-type DCs treated with DMSO (gray) or NSC23766 (blue) as in (**a**). n = 6. **d.** Wild-type DCs migrating treated with DMSO (left) or NSC23766 (right) migrating under a patch of 1.0% agarose fixed and stained with phalloidin (F-actin, black) and DAPI (nucleus, blue). **e.** Dots show percentages of DMSO or NSC23766 treated DCs showing a central actin pool, in two independent experiments. DMSO, n = 90; NSC, n = 68. ns P > 0.9999. **f.** Wild-type DCs treated with DMSO (left) or ML141 (right) as in (**d**). **g.** Dots show percentages of DMSO or ML141 treated DCs as in (**e**) in two independent

experiments. DMSO, n = 66; ML141, n = 78. **** P > 0.0001. **h.** Central actin pool was segmented based on its phalloidin intensity (left). Mean values obtained in the central actin pool region were normalized to overall F-actin intensity in the cell (right). **i.** Wild-type DCs expressing GFP, Cdc42^WT-GFP or Cdc42^T17N-GFP as in (**d**). **j.** Dots show percentage of GFP+, Cdc42^WT-GFP+ or Cdc42^T17N-GFP+ DCs as in (**e**) in three independent experiments. GFP, n = 32; WT, n = 27; T17N, n = 52. ns P = 0.6780, ** P = 0.0018, *** P = 0.0004. **k.** Wild-type and *Wasp*^−/− DCs as in (**d**). **l.** Dots show percentage of wild-type and *Wasp*^−/− DCs as in (**e**) in two independent experiments. WT, n = 240; *Wasp*^−/−, n = 149. **** P < 0.0001. Histogram bars (**e, g, j, l**) are mean ± s.e.m. and error bars in (**a–c**) show the s.e.m. Two-sided Fisher's exact test is used in (**e, g, j, l**). Scale bars in (**d, f, i, k**) are 10 μm. n indicates number of movies in (**a–c**) and number of cells in (**e, g, j, l**).

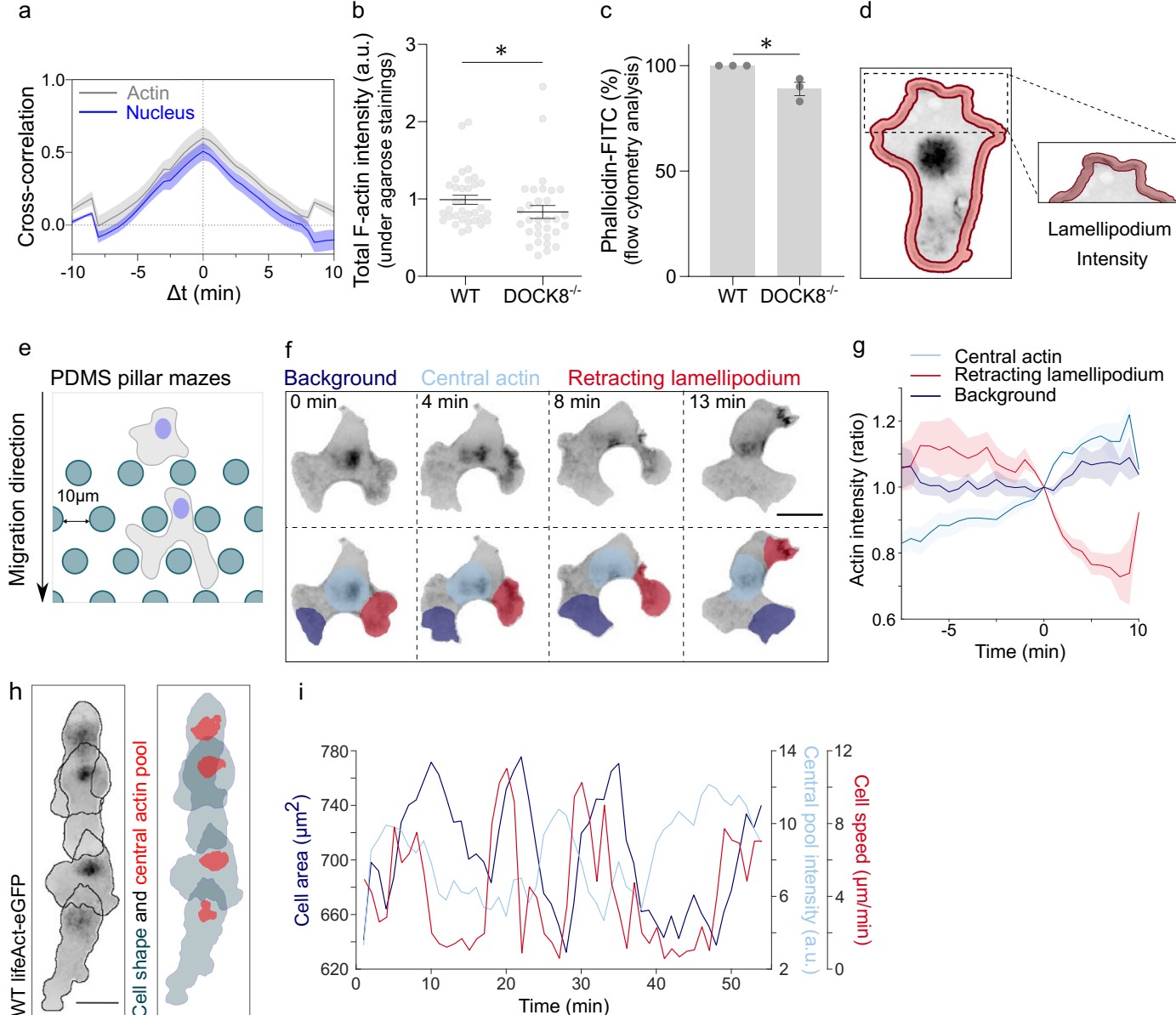

**Extended Data Fig. 4 | Extended Data Fig. 4 related to Fig. 4. a.** Temporal cross-correlation between bead displacement and nucleus (blue) or actin intensity (grey) in *Dock8*⁻/⁻ DCs migrating under agarose mixed with fluorescent beads. n = 17 **b**. Dots show mean total F-actin intensity (phalloidin) of wild-type and *Dock8*⁻/⁻ fixed DCs migrating under 1.0% agarose in three independent experiments. WT, n = 34; *Dock8*⁻/⁻, n = 26. * P < 0.05 Two-tailed paired t-test. **c.** Dots show normalized total F-actin (phalloidin-FITC) intensity accessed by flow cytometry of wild-type and *Dock8*⁻/⁻ DCs in three independent experiments. * P = 0.03808. Two-sided Fisher's exact test **d.** Scheme showing cell segmentation for quantification of lamellipodial actin. **e.** Scheme showing DCs migrating in a PDMS pillar maze. Cells were confined between 6 μm apart surfaces intersected by 10 μm distanced pillars. **f.** Top: Time-lapse images of a LifeAct-eGFP⁺ (actin, black)

DC migrating in a pillar maze as shown in (**e**). Bottom: Regions used for quantification of the retracting lamellipodium (red), central actin pool (light blue), and an area where no substantial changes in actin intensity was observed - background (dark blue). **g.** Normalized mean actin intensity (LifeAct-eGFP) in the central pool (light blue), the retracting lamellipodium (red), and background (dark blue) through time. n = 24. **h.** Left: Time-lapse projection of a migrating LifeAct-eGFP⁺ (actin, black) DC. Black line shows cell contour. Right: DC shape (blue) and central actin pool (red) segmentation results. **i.** Cell area (dark blue), central actin pool intensity (light blue) and cell speed (red) changes of a single DC migrating under 1.0% agarose over time. Histogram bars (**c**) are mean ± s.e.m., error bars in (**b**) show the s.e.m and s.d. in (**g**). Scale bars in (**f**, **h**) are 15 μm. n indicates number of cells.

# Reporting Summary

## Statistics

For all statistical analyses, confirm that the following items are present in the figure legend, table legend, main text, or Methods section.

| n/a | Confirmed | |
|---|---|---|
| ☐ | ☒ | The exact sample size (*n*) for each experimental group/condition, given as a discrete number and unit of measurement |
| ☐ | ☒ | A statement on whether measurements were taken from distinct samples or whether the same sample was measured repeatedly |
| ☐ | ☒ | The statistical test(s) used AND whether they are one- or two-sided <br> *Only common tests should be described solely by name; describe more complex techniques in the Methods section.* |
| ☒ | ☐ | A description of all covariates tested |
| ☒ | ☐ | A description of any assumptions or corrections, such as tests of normality and adjustment for multiple comparisons |
| ☐ | ☒ | A full description of the statistical parameters including central tendency (e.g. means) or other basic estimates (e.g. regression coefficient) AND variation (e.g. standard deviation) or associated estimates of uncertainty (e.g. confidence intervals) |
| ☐ | ☒ | For null hypothesis testing, the test statistic (e.g. *F*, *t*, *r*) with confidence intervals, effect sizes, degrees of freedom and *P* value noted <br> *Give P values as exact values whenever suitable.* |
| ☒ | ☐ | For Bayesian analysis, information on the choice of priors and Markov chain Monte Carlo settings |
| ☒ | ☐ | For hierarchical and complex designs, identification of the appropriate level for tests and full reporting of outcomes |
| ☒ | ☐ | Estimates of effect sizes (e.g. Cohen's *d*, Pearson's *r*), indicating how they were calculated |

*Our web collection on statistics for biologists contains articles on many of the points above.*

## Software and code

Policy information about availability of computer code

| Data collection | Imaging data was collected with: NIS Elements 5.3 (Nikon Instruments - Nikon TiE2); ZEN 3.8 (LSM800 and LSM800 inverted, Zeiss); Nikon JOBS v.5.02 and NIS Elements 5.3 (Nikon CSU-W1); Fusion 2.2 and Imaris v9.91 (Andor Dragonfly 505); VisiView software (Visitron, Zeiss Axio Observer.Z1 inverted); FACS Diva 6.1.3 (FACS Canto BD Biosciences), CytExpert (BC CytoFLEX LX); Cell Sorter Software V2.24 (Sony SH800 SFP sorter). |
|---|---|
| Data analysis | Fiji (ImageJ) v1.52; Ilastik v1.4; Matlab R2020a; Python v3.9; Davis 8 (Lavision); GraphPad Prism v10. <br> All custom-made scripts used are available upon request. |

For manuscripts utilizing custom algorithms or software that are central to the research but not yet described in published literature, software must be made available to editors and reviewers. We strongly encourage code deposition in a community repository (e.g. GitHub). See the Nature Portfolio guidelines for submitting code & software for further information.

## Data

Policy information about availability of data

All manuscripts must include a data availability statement. This statement should provide the following information, where applicable:

- Accession codes, unique identifiers, or web links for publicly available datasets
- A description of any restrictions on data availability
- For clinical datasets or third party data, please ensure that the statement adheres to our policy

> All data supporting the fidings in this study are available upon request.

## Research involving human participants, their data, or biological material

Policy information about studies with human participants or human data. See also policy information about sex, gender (identity/presentation), and sexual orientation and race, ethnicity and racism.

| | |
|---|---|
| Reporting on sex and gender | *Use the terms sex (biological attribute) and gender (shaped by social and cultural circumstances) carefully in order to avoid confusing both terms. Indicate if findings apply to only one sex or gender; describe whether sex and gender were considered in study design; whether sex and/or gender was determined based on self-reporting or assigned and methods used. Provide in the source data disaggregated sex and gender data, where this information has been collected, and if consent has been obtained for sharing of individual-level data; provide overall numbers in this Reporting Summary. Please state if this information has not been collected. Report sex- and gender-based analyses where performed, justify reasons for lack of sex- and gender-based analysis.* |
| Reporting on race, ethnicity, or other socially relevant groupings | *Please specify the socially constructed or socially relevant categorization variable(s) used in your manuscript and explain why they were used. Please note that such variables should not be used as proxies for other socially constructed/relevant variables (for example, race or ethnicity should not be used as a proxy for socioeconomic status). Provide clear definitions of the relevant terms used, how they were provided (by the participants/respondents, the researchers, or third parties), and the method(s) used to classify people into the different categories (e.g. self-report, census or administrative data, social media data, etc.) Please provide details about how you controlled for confounding variables in your analyses.* |
| Population characteristics | *Describe the covariate-relevant population characteristics of the human research participants (e.g. age, genotypic information, past and current diagnosis and treatment categories). If you filled out the behavioural & social sciences study design questions and have nothing to add here, write "See above."* |
| Recruitment | *Describe how participants were recruited. Outline any potential self-selection bias or other biases that may be present and how these are likely to impact results.* |
| Ethics oversight | *Identify the organization(s) that approved the study protocol.* |

Note that full information on the approval of the study protocol must also be provided in the manuscript.

# Field-specific reporting

Please select the one below that is the best fit for your research. If you are not sure, read the appropriate sections before making your selection.

☒ Life sciences ☐ Behavioural & social sciences ☐ Ecological, evolutionary & environmental sciences

For a reference copy of the document with all sections, see nature.com/documents/nr-reporting-summary-flat.pdf

# Life sciences study design

All studies must disclose on these points even when the disclosure is negative.

| | |
|---|---|
| Sample size | No statistical methods were used to predetermine sample sizes. Sample sizes were chosen based on previosu experience and published studies to assess reproducibility. Experiments were repeated multiple times and included parallel imaging of multiple cells, resulting in sample sizes in the range of tens to hundreds. Every migratory cell was analysed. |
| Data exclusions | To avoid artifacts in cell speed and cell area, only single cell tracks of cells migrating under agarose or microfluidic PDMS devices were included. For speed analysis in PDMS straight channels, only single cells migrating in the channel were considered. |
| Replication | When possible, analysis was performed with at least twice entirely independent experiments (e.g. independent dendritic cell differentiations, multiple individual cells per individual experiments, and handled at different days). All attempts at replication were successful. |
| Randomization | Cells were randomized into control and experimental groups. |
| Blinding | Experiments and data analysis were not blinded. Analysis of subjective data, such as MTOC-nucleus or central actin-nucleus positioning, was |

| Blinding | performed by two different individuals with reproducible results. |

# Reporting for specific materials, systems and methods

We require information from authors about some types of materials, experimental systems and methods used in many studies. Here, indicate whether each material, system or method listed is relevant to your study. If you are not sure if a list item applies to your research, read the appropriate section before selecting a response.

## Materials & experimental systems

| n/a | Involved in the study |
|---|---|
| ☐ | ☒ Antibodies |
| ☐ | ☒ Eukaryotic cell lines |
| ☒ | ☐ Palaeontology and archaeology |
| ☐ | ☒ Animals and other organisms |
| ☒ | ☐ Clinical data |
| ☒ | ☐ Dual use research of concern |
| ☒ | ☐ Plants |

## Methods

| n/a | Involved in the study |
|---|---|
| ☒ | ☐ ChIP-seq |
| ☐ | ☒ Flow cytometry |
| ☒ | ☐ MRI-based neuroimaging |

## Antibodies

| Antibodies used | Primary antibodies used (Reagent, Species, Clonality, Conjugate, Dilution, Source, Cat. No): |
|---|---|

anti-MHC II, rat IgG, Monoclonal M5/114.15.2, eFluor450, 1:400, ebioscience, 48-5321-82;
anti-CD11c, armenian hamster/ IgG, Monoclonal N418, APC, 1:150, ebioscience, 17-0114-82;
anti-CD16/32, rat/IgG2a lambda, Monoclonal clone 93, unconjugated, 1:100, ebioscience,14-0161-85;
anti-gamma-2-tubulin, rabbit IgG, polyclonal, unconjugated, 1:400, abcam, ab11317;
anti-Giantin, rabbit, polyclonal, unconjugated, 1:100, Sysy antibodies, 263003;
anti-LAMP2, rat IgG, monoclonal, unconjugated, 1:100, abcam, ab13524;
anti-Talin, mouse, monoclonal clone 8d4, unconjugated, 1:400, Sigma T3287;
anti-HSPA1A (HSP70), rabbit IgG, polyclonal, unconjugated, 1:10 000, Thermofisher Scientific, PA5-34772.

Secondary antibodies (Reagent, Species, Clonality, Conjugate, Dilution, Source, Cat. No):

anti-rabbit, goat IgG (H+L), polyclonal, Alexa Fluor-488, 1:200-1:400, Invitrogen, A-11008;
anti-rat, donkey  IgG (H+L), polyclonal, Alexa FLuor-488, 1:200-1:400, Jackson Immuno Research, AB_2340686;
anti-mouse, goat IgG (H+L), monoclonal, HRP conjugate, 1:10 000, BioRad, 1706516;
anti-rabbit, goat IgG (H+L), monoclonal, HRP conjugate, 1:3 000, BioRad, 1706515.

Other reagents used for staining:
Alexa Flour647 Phalloidin,  1:400, Invitrogen, A22287;
Fluorescein (FITC) Phalloidin, 1:200, Invitrogen, F432.

| Validation | All antibodies are commercial standard validated antibodies. Validation data is available on vendor websites. Antibodies were tested using known positive and negative controls and titrated following the manufacturer's recommendations. |
|---|---|

## Eukaryotic cell lines

Policy information about cell lines and Sex and Gender in Research

| Cell line source(s) | Lenti-X-293 derived from HEK 293 cells (TakaraBio); Dendritic cells and T cells were obtained from primary cell cultures as described in the Methods section. |
|---|---|
| Authentication | Primary cell lines have been tested by antibody markers for differentiation in the respective cell types. |
| Mycoplasma contamination | All cell lines were tested negative for mycoplasma contamination. |
| Commonly misidentified lines (See ICLAC register) | No commonly missindentified cell lines were used. |

## Animals and other research organisms

Policy information about studies involving animals; ARRIVE guidelines recommended for reporting animal research, and Sex and Gender in Research

| Laboratory animals | Laboratory mice (mus musculus) from the following backgrounds were used: WT C57BL/6 (Janvier), WASp-/- ( B6.129S6-Wastm1Sbs/J; No. 019458; The Jackson Laboratory) and DOCK8-/- a gift from Yoshinori Fukui's lab). All mice used in this study were bred on a C57BL/6 background and maintained at the Institute of Science and Technology Austria (ISTA) institutional animal facility following |
|---|---|

the guidelines from its ethics commission and the Austrian law for animal experimentation. Mice with 8 to 12 weeks of age were used for organ removal and cell extraction.

| | |
|---|---|
| Wild animals | This study did not involve wild animals. |
| Reporting on sex | Both male and female animals indistinctly used in this study. |
| Field-collected samples | This study did not involve samples collected from the field. |
| Ethics oversight | Guidelines from the ethics commission of ISTA's animal facility and the Austrian law for animal experimentation were followed during the course of this study. |

Note that full information on the approval of the study protocol must also be provided in the manuscript.

## Plants

| | |
|---|---|
| Seed stocks | *Report on the source of all seed stocks or other plant material used. If applicable, state the seed stock centre and catalogue number. If plant specimens were collected from the field, describe the collection location, date and sampling procedures.* |
| Novel plant genotypes | *Describe the methods by which all novel plant genotypes were produced. This includes those generated by transgenic approaches, gene editing, chemical/radiation-based mutagenesis and hybridization. For transgenic lines, describe the transformation method, the number of independent lines analyzed and the generation upon which experiments were performed. For gene-edited lines, describe the editor used, the endogenous sequence targeted for editing, the targeting guide RNA sequence (if applicable) and how the editor was applied.* |
| Authentication | *Describe any authentication procedures for each seed stock used or novel genotype generated. Describe any experiments used to assess the effect of a mutation and, where applicable, how potential secondary effects (e.g. second site T-DNA insertions, mosiacism, off-target gene editing) were examined.* |

## Flow Cytometry

### Plots

Confirm that:

☐ The axis labels state the marker and fluorochrome used (e.g. CD4-FITC).

☐ The axis scales are clearly visible. Include numbers along axes only for bottom left plot of group (a 'group' is an analysis of identical markers).

☐ All plots are contour plots with outliers or pseudocolor plots.

☒ A numerical value for number of cells or percentage (with statistics) is provided.

### Methodology

| | |
|---|---|
| Sample preparation | Cell suspension was filtered to avoid cell aggregates.<br>For analysis of surface expression markers in DCs, cells were counted and equal numbers of cells were stained in parallel with different antibodies. |
| Instrument | Data on surface expression markers was acquired either in a FACS Canto BD Biosciences or in a BC CytoFLEX LX machine.<br>Data on progenitor cell line reporters was acquired in a Sony SH800 FP cell sorter (sorting chip: 100 um). |
| Software | Analysis of surface expression markers was performed using either FACS Diva 6.1.3 (FACS Canto BD Biosciences) or CytExpert software (BD CytoFLEX LX). Analysis of positive progenitor cell line reporters was performed using Cell Sorter Software V2.24. |
| Cell population abundance | For establishment of progenitor cell line reporters a minimum of 500 000 positive cells (expressing either GFP or mCherry) were sorted. |
| Gating strategy | Doublets were excluded based on FSC and SSC profile.<br>For the establishment of progenitor cell line reporters, cells were discriminated based on the signal distribution in a single fluorescence channel. |

☐ Tick this box to confirm that a figure exemplifying the gating strategy is provided in the Supplementary Information.

