## [Peer Review File · Nature Immunology]

Migrating immune cells globally coordinate protrusive forces

Corresponding Author: Professor Michael Sixt

Version 0:

Decision Letter:

5th Aug 2024

Dear Dr. Sixt,

Your Letter, "Global coordination of protrusive forces in migrating immune cells" has now been seen by 2 referees. While we find your work of considerable potential interest, the reviewers have raised concerns that must be addressed. As such, we cannot accept the current version of the manuscript for publication, but would be happy to consider a revised version that addresses these concerns, as long as novelty is not compromised in the interim.

Please revise the manuscript to address all issues raised by the referees. At resubmission, please include a point-by-point "Response to referees" detailing how you have addressed each referee comment (please specify the page and figure number where the new data can be found in the revised manuscript and please highlight the changes in the manuscript as well). This response will be sent back to the referees along with the revised manuscript.

In addition, please include a revised version of any required reporting checklist. It will be available to referees (and, potentially, statisticians) to aid in their evaluation if the manuscript goes back for peer review. A revised checklist is essential for re-review of the paper. The Reporting Summary can be found here: <https://www.nature.com/documents/nr-reporting-summary.pdf>

When submitting the revised version of your manuscript, please pay close attention to our <https://www.nature.com/nature-portfolio/editorial-policies/image-integrity> Digital Image Integrity Guidelines. and to the following points below:

Extended Data figures and tables are online-only (appearing in the online PDF and full-text HTML version of the paper), peer-reviewed display items that provide essential background to the Article but are not included in the printed version of the paper due to space constraints or being of interest only to a few specialists. A maximum of ten Extended Data display items (figures and tables) is typically permitted. When re-submitting your manuscript, please ensure that any supplementary figures and tables that are more critical to the manuscript's conclusions are converted to Extended data to increase these data's visibility.

Link Redacted

We hope to receive a suitably revised manuscript within 4 months. If you cannot send it within this time, please let us know. We will be happy to consider your revision so long as nothing similar has been accepted for publication at Nature Immunology or published elsewhere.

Nature Immunology is committed to improving transparency in authorship. As part of our efforts in this direction, we are now requesting that all authors identified as 'corresponding author' on published papers create and link their Open Researcher and Contributor Identifier (ORCID) with their account on the Manuscript Tracking System (MTS), prior to acceptance. ORCID helps the scientific community achieve unambiguous attribution of all scholarly contributions. You can create and link your ORCID from the home page of the MTS by clicking on 'Modify my Springer Nature account'. For more information please visit www.springernature.com/orcid.

Thank you for the opportunity to review your work.

Sincerely,

Ioana Staicu, Ph.D.
Senior Editor
Nature Immunology

Tel: 212-726-9207
Fax: 212-696-9752
www.nature.com/ni

Reviewers' Comments:

Reviewer #1:

Remarks to the Author:

Reis-Rodrigues et al

This study focused on mechanisms through which dendritic cells adapt to migrate in tissue environments with different pore size and stiffnesses. The authors show that normal DC that face obstructed environments can switch from amoeboid locomotion with the nucleus in front to mesenchymal migration with the nucleus trailing. This transition involves generation of a central F-actin pool that enables the cell to generate orthogonal forces to generate openings sufficiently soft environments to make forward progress along a chemokine gradient. They use a clever assay putting beads in an agarose layer over the cells to track μm scale displacements as the cell moves under the agarose. They find that this central F-actin pool is Cdc42 and Dock8 dependent. The ability to form this central F-actin pool both enables the orthogonal forces and restrains the leading edge to promote cell integrity- not leaving the nucleus behind. There are a number of very interesting mechanistic steps and concepts in this study. I have a few questions to help understand the amoeboid-mesenchymal analogy applied to these cells.

1. As discussed by the authors, mesenchymal migration strategies generally involve adhesion to pull cells forward against resistance. The authors have also previously shown that normal DC migration from the periphery is almost entirely integrin independent. Does this just mean that migration from sites such as skin lack any restrictions that would require the switch to the mesenchymal strategy, or that the implementation of this strategy by DC doesn't require adhesion and is thus on this spectrum, but not equivalent to how a mesenchymal cell would migrate? Experiments to verify the role or lack of role for integrins or other adhesion systems would be valuable.
2. Can the authors confirm that the CDC42 inhibitor treated or Dock8 deficient DC lose the ability to vertically displace the agarose during under agarose migration? This is a prediction that is not tested, but is only presented as a model in Figure 4.
3. How does the central F-actin pool generate protrusive force? Is it using the microtubule network as a rigid skeleton or some other scaffold. This seems like one of the most interesting problems. It would be somewhat reminiscent of the tensegrity model. Or is this the mechanism more similar to a podosome like actin-myosin anchoring the the surrounding plasma membrane, which would again invoke adhesion.

Reviewer #2:

Remarks to the Author:

In this superb manuscript, Reis-Rodrigues et al. probe how immune cells maintain the coherence of their migration in complex 3D environments. They find that cell compression stimulates the formation of a central actin pool. This pool serves two functions: (1) to locally expand a path for the nucleus and (2) to compete with the protrusive program to limit cell advance when the trailing edge is restricted. The authors identify a core component of this central actin pool Dock8 that they use to demonstrate the importance of this structure for cell migration in complex environments. This paper represents a significant advance in our understanding of how cells maintain coherence in migration. The experiments are creative and compelling, and the writing is clear. I have a few suggestions for buttressing the core findings.

Major points

1. The authors identify one protein that is essential for the central actin pool (Dock8) and one that plays a significant role

(Cdc42). They previously identified a role for WASP in generating space for lamellipodial advance in compressive environments (Gaertner et al., 2022), and though it wasn't the focus of this paper, it appears to play an important role in the central actin pool as well. It would be valuable to assemble these proteins into a pathway (likely Dock8 to Cdc42 to WASP, but possibly more complex if there is feedback) by examining the localization of each component in the absence of the other. Similarly, for the core component they have focused on here (Dock8), it would be useful to examine whether it is permissive or instructive for the central actin pool by examining Dock8 dynamics as cells navigate constrictive barriers.

2. The authors find a negative correlation between the central actin pool and cell protrusions. This suggests potential competition between these programs, which is supported by enhanced protrusion when the central actin pool is inhibited. The simplest interpretation is that these two structures compete for actin. To help buttress this point, it would be useful to do a more careful accounting of actin polymer to determine if the amount gained in central actin and lost in a given retracting protrusion are equivalent, and if the timing is consistent with this competition.

3. The cell fragmentation in complex environments is a striking phenotype. It is not obvious whether hyperprotrusion alone can explain it, or if it requires a change in cortical properties of the cell—for example the loss of cell coherence that accompanies myosin inhibition in some cells. On this note, it would be interesting to examine whether myosin activation is misregulated in the Dock8 null cells.

4. Is this central actin pool upon confinement unique to dendritic cells, or is it a more general feature of immune cells? From the author's previous work, it seems less apparent in lymphocytes, but it would be nice to have a direct comparison with other immune cells in the current manuscript. I'd be satisfied with just the LifeAct experiments.

5. What is the relation between the central actin pool and bead displacement? For Fig 2c, it appears that the bulk of the bead displacement occurs before the central actin pool. Could the authors comment on this? I would like to see the same plot in Dock8 null cells. What about the rare cells where the central actin pool lags the nucleus? Or the subset of Cdc42 inhibited cells with and without a central actin pool? These experiments are important to establish the role of the central actin pool in vertical forces.

Minor points

1. I don't see titles for the figures in the figure legends. Their addition would be useful to help understand what is in each.

2. Now that there are much better tools, the authors shouldn't be using dominant negative Cdc42. Because this acts by sequestering upstream GEFs, any other GTPases that share this GEF will also be inhibited, and the literature abounds with artifacts from dominant negative GTPases. Knockdown or knockout of Cdc42 would be a much better approach.

3. For cross-correlations, use seconds, not frames for the x-axis.

4. The framing of mesenchymal vs ameboid configuration is likely to be confusing to readers that may assume the cells are fully adopting a mesenchymal migration mode, which has many other features than just position of the nucleus. I'd recommend just talking about nuclear position instead of this mesenchymal framework.

5. The authors propose that the central actin pool may protect the nucleus. Can they see evidence in changes in nuclear deformation (in agarose or collagen or microfluidic chambers) with and without the central actin pool?

6. Line 165: "WASP-dependent"

7. Line 737: "MTOC-first" or "MTOC first"?; standardize hyphen usage throughout manuscript

8. Line 746: "DCs"

9. Line 810: "AF555 (red)"

10. Fig. 4g: Being able to see differences in speed very difficult to observe in Supp. Video 5 and only indicated by this cross-correlation plot – any way to depict these changes in speed in the video?

11. Extended Data Fig. 4e: Background segmentation ("area where no sig. change in cell morph. was observed", Line 973) seems to undergo large shape changes in panels and in Supp Vid 5; can the authors explain how background was chosen in more detail?

Version 1:

Decision Letter:

Our ref: NI-LE38275A

22nd Apr 2025

Dear Dr. Sixt,

Thank you for submitting your revised manuscript "Global coordination of protrusive forces in migrating immune cells" (NI-LE38275A). It has now been seen by the original referees and their comments are below. We are happy to inform you that if you revise your manuscript appropriately according to our editorial requirements, your manuscript should be publishable in Nature Immunology.

I will now pre-edit the current version of your paper. We will also perform detailed checks on your paper and will send you a checklist detailing our editorial and formatting requirements in about two weeks. Please do not upload the final materials and make any revisions until you receive this additional information from us.

While waiting for the pre-edit check, please deposit all omic and code data into public repositories so that the accession codes are readily available to be added in the revised manuscript. We cannot accept the paper without the codes.

In addition, all corresponding authors need to update and link their ORCID to their Nature account. We cannot accept the paper without this information. Should you have any query or comments about ORCID, please do not hesitate to contact our editorial assistant at immunology@us.nature.com.

If you had not uploaded a Word file for the current version of the manuscript, we will need one before beginning the editing process; please email that to immunology@us.nature.com at your earliest convenience.

Thank you again for your interest in Nature Immunology. Please do not hesitate to contact me if you have any questions.

Sincerely,

Ioana Staicu, Ph.D.
Senior Editor
Nature Immunology

Tel: 212-726-9207
Fax: 212-696-9752
www.nature.com/ni

Reviewer #1 (Remarks to the Author):

The authors have addressed my concerns. I agree with holding off on discussion of the microbutubule effects, but appreciate the authors sharing the preliminary result.

Reviewer #2 (Remarks to the Author):

The authors have satisfactorily addressed all of my concerns, and I strongly recommend publication of this innovative important manuscript.

A quick aside to the authors-- from their rebuttal, I know that the microtubule perturbations are not going to be a part of this paper, but for their future research in this space, I'd recommend the use of vinblastine to assay the role of microtubules. Nocodazole can be complicated, as it releases RhoGEFs by depolymerizing microtubules, hyperactivating Rho and myosin-based contraction. this makes it difficult to discern a role for microtubule mass vs microtubules themselves in actin-based processes. Vinblastine is a much cleaner perturbation to assay the roles of microtubules in actin-based processes (see for example PMID: 16176947).

Reviewers' Comments

Reviewer #1:

Remarks to the Author:

Reis-Rodrigues et al

This study focused on mechanisms through which dendritic cells adapt to migrate in tissue environments with different pore size and stiffnesses. The authors show that normal DC that face obstructed environments can switch from amoeboid locomotion with the nucleus in front to mesenchymal migration with the nucleus trailing. This transition involves generation of a central F-actin pool that enables the cell to generate orthogonal forces to generate openings sufficiently soft environments to make forward progress along a chemokine gradient. They use a clever assay putting beads in an agarose layer over the cells to track μm scale displacements as the cell moves under the agarose. They find that this central F-actin pool is Cdc42 and Dock8 dependent. The ability to form this central F-actin pool both enables the orthogonal forces and restrains the leading edge to promote cell integrity- not leaving the nucleus behind. There are a number of very interesting mechanistic steps and concepts in this study. I have a few questions to help understand the amoeboid-mesenchymal analogy applied to these cells.

1. As discussed by the authors, mesenchymal migration strategies generally involve adhesion to pull cells forward against resistance. The authors have also previously shown that normal DC migration from the periphery is almost entirely integrin independent. Does this just mean that migration from sites such as skin lack any restrictions that would require the switch to the mesenchymal strategy, or that the implementation of this strategy by DC doesn't require adhesion and is thus on this spectrum, but not equivalent to how a mesenchymal cell would migrate? Experiments to verify the role or lack of role for integrins or other adhesion systems would be valuable.

Following the reviewer's suggestion, we investigated the role of integrin mediated adhesions in DC polar organization as well as formation of the central actin pool. We generated Talin 1 knock out DCs (TLN^{-/-}, **Extended Data Fig. 1a, b**) and tested them in collagen gels of varying concentrations. Like WT DCs, TLN^{-/-} cells predominantly migrated nucleus-first in low density collagen gels, and switched to a MTOC-first configuration in high density collagen gels (**Fig. 1b-e**). We also detected a central actin pool in TLN^{-/-} DCs expressing LifeAct-eGFP migrating under agarose (**Supplementary Video 3, Third part**) and the same was true when WT DCs migrated on PEG passivated substrates (**Supplementary Video 3, Second part**). Accordingly, no adhesion markers localized to the central actin pool in WT DCs (**Supplementary Video 3, First part**). These results suggest that adhesions are neither required for the formation of the central actin pool, nor necessary for organelle re-orientation upon passage through tight pores. In the revised version of this manuscript, we now decided to use the term "organelle re-orientation" instead of "amoeboid to mesenchymal transition".

2. Can the authors confirm that the CDC42 inhibitor treated or Dock8 deficient DC lose the ability to vertically displace the agarose during under agarose migration? This is a prediction that is not tested, but is only presented as a model in Figure 4.

We have updated the text and Fig. 4 to address this point. Specifically, we have added snap-shots and a video of a representative DOCK8^{-/-} DC moving under-agarose with fluorescent beads (**Fig. 4b and Supplementary Video 11**) and show the corresponding change in bead displacement in **Fig. 4d**. Cross-correlation between bead displacement and either LifeAct-eGFP or Hoechst intensities can now be found in **Extended Data Fig. 4a**. In summary, the data show that DOCK8^{-/-} DCs impose smaller deformations on the agarose. As the central actin pool is absent in these cells, the remaining deformations are associated with the nucleus.

3. How does the central F-actin pool generate protrusive force? Is it using the microbutule network as a rigid skeleton or some other scaffold. This seems like one of the most interesting problems. It would be somewhat reminiscent of the tensegrity model.

We agree that this is an interesting open question. Treatment of DCs with doses of nocodazole that resulted in complete loss of the microtubule network (**Additional Data Fig. 1a**) indeed caused a decisive decrease of the percentage of cells showing the central actin pool (**Additional Data Fig. 1d**). These data demonstrate that the microtubule network does play a role in maintenance and localization of the central actin pool and might even indicate that microtubules serve a load-bearing function as suggested by the reviewer.

However, we are hesitant to add these data to the manuscript as they are difficult to interpret. In DCs the centrosome acts as the sole microtubule organizing center (MTOC) and lysosomes and Golgi apparatus localize close to the MTOC, where they are presumably transported by minus end motors (**Extended Data Fig. 1i**). Accordingly, Nocodazole treatment caused substantial dispersion of lysosomes and Golgi apparatus throughout the cell body (**Additional Data Fig. 1b, c**). Hence, the loss of the central actin pool could also be an indirect consequence of organelle displacement. A recent publication reported the presence of DOCK8 in the lysosomal membrane where it controls actin polymerization (Gutierrez-Ruiz et al Cell Rep, 2023), suggesting that DOCK8 on endomembranes might trigger the central actin pool. We think that disentangling the role of microtubules in force generation vs. organelle positioning will require extensive and thorough follow up investigations that are beyond the scope of our current work. We therefore share these results with the reviewers but did not include them into the manuscript.

Additional Data Figure 1. Nocodazole treatment leads to organelle dispersion and consequent loss of the central actin pool. **a.** Representative images of DMSO (control) or Nocodazole treated (300 nM) DCs moving under 1% agarose stained with: **a.** Phalloidin (actin, red) and α -Tubulin (microtubules, cyan); **b.** Phalloidin (actin, red) and LAMP2A (lysosomes, cyan); and **c.** Phalloidin (actin, red) and Giantin (Golgi complex, cyan). Nucleus position in all images is shown by the white overlay. Scale bar, 10 μ m. **d.** Cells showing a central pool of actin upon treatment with DMSO or Nocodazole (300 nM, Noco.). Each dot represents the percentage of cells where a central actin pool was detectable in three independent experiments. DMSO treated cells, n=73; Nocodazole treated cells, n=100. Error bars show the Standard Error of the Mean (SEM). **** p-value<0.0001. Fisher's exact test.

Or is this the mechanism more similar to a podosome like actin-mysin anchoring the surrounding plasma membrane, which would again invoke adhesion.

Following the suggestion of the reviewer, we addressed the role of myosin II in the central actin pool. First, we revisited data from our previous manuscript on WASp dependent actin patches that generate local deformations in the surrounding environment of a migrating leukocyte (Gartner et al, 2020). Here we observed that myosin inhibition with para-Nitroblebbistatin did neither affect actin accumulation at the lamellipodium (Gartner et al, 2020) nor at the central actin pool (**Additional Data Fig. 2 a**). Additionally, we imaged myosin light-chain (MLC)-mKate expressing DCs under agarose and detected the typical accumulation of MLC at the back (**left and middle panel**) and at regions of protrusion retraction (**right panel, Additional Data Fig. 2 b**), but not at the region of the central actin pool. Together, these data suggest that myosin does not play a role in the force generated in this region.

Additional Data Figure 2. a. LifeAct-eGFP expressing DCs (black, actin) migrating in PEG coated coverslips under-agarose. Cells were treated with DMSO (control, top) or para-Nitroblebbistatin (100 μ M, bottom). Note that the presence of the central actin pool is independent of Myosin II activity. Scale bar, 15 μ m. **b.** Time lapse of a Myosin Light Chain (MLC)-mKate expressing DC migrating under 1% agarose. Red arrows highlight MLC accumulation at the back of the cell (left and middle panel), or accumulation at places of protrusion retraction (right panel). Note that we observed no myosin accumulation at the central actin pool region. Scale bar, 10 μ m.

Reviewer #2:

Remarks to the Author:

In this superb manuscript, Reis-Rodrigues et al. probe how immune cells maintain the coherence of their migration in complex 3D environments. They find that cell compression stimulates the formation of a central actin pool. This pool serves two functions: (1) to locally expand a path for the nucleus and (2) to compete with the protrusive program to limit cell advance when the trailing edge is restricted. The authors identify a core component of this central actin pool Dock8 that they use to demonstrate the importance of this structure for cell migration in complex environments. This paper represents a significant advance in our understanding of how cells maintain coherence in migration. The experiments are creative and compelling, and the writing is clear. I have a few suggestions for buttressing the core findings.

Major points

1. The authors identify one protein that is essential for the central actin pool (Dock8) and one that plays a significant role (Cdc42). They previously identified a role for WASP in generating space for lamellipodial advance in compressive environments (Gaertner et al., 2022), and though it wasn't the focus of this paper, it appears to play an important role in the central actin pool as well. It would be valuable to assemble these proteins into a pathway (likely Dock8 to Cdc42 to WASP, but possibly more complex if there is feedback) by examining the localization of each component in the absence of the other.

Indeed the exact interplay between DOCK8, Cdc42 and their effectors remains an interesting yet highly complex open question. As proposed by the reviewer, we investigated the pathway below Cdc42 by analyzing eGFP-WASp in DCs migrating under agarose. As previously shown (Gaertner et al, Dev Cell, 2022) WASp localized not only as small patches at the periphery of cells, but also at their

center at the region of the central actin pool (**Supplementary Video 9, First part**). However, analysis of WASp depleted DCs revealed only a mild decrease in the prevalence of cells showing the central actin pool (**Extended Data Fig. 3k, l**). Interestingly, in DOCK8 depleted cells, eGFP-WASp localization at the periphery was retained, but completely lost from the central actin pool (**Supplementary Video 9, Second part**). We also attempted to overexpress DOCK8-GFP in both WT and WASp depleted cells, but it was unfortunately lethal in our system. Overall, our results show that WASp, downstream of DOCK8 and Cdc42, contributes but it is not essential for the generation of the central actin pool, suggesting the participation of other Cdc42 effectors that remain to be uncovered.

Similarly, for the core component they have focused on here (Dock8), it would be useful to examine whether it is permissive or instructive for the central actin pool by examining Dock8 dynamics as cells navigate constrictive barriers.

As suggested, we now examined DOCK8 dynamics on the DOCK8 depleted cells rescued with DOCK8-GFP (**Fig. 3g, h**). We let these cells pass through constricted channels and observed a striking accumulation of DOCK8 at the site of constriction (**Fig. 3i, j and Supplementary Video 10**). This demonstrates that DOCK8 accumulation precisely replicates the actin accumulation previously observed (**Fig. 1i and Supplementary Video 2, Second part**). We think this spatio-temporal association of DOCK8 and actin polymerization argues for an instructive role.

2. The authors find a negative correlation between the central actin pool and cell protrusions. This suggests potential competition between these programs, which is supported by enhanced protrusion when the central actin pool is inhibited. The simplest interpretation is that these two structures compete for actin. To help buttress this point, it would be useful to do a more careful accounting of actin polymer to determine if the amount gained in central actin and lost in a given retracting protrusion are equivalent, and if the timing is consistent with this competition.

In fission yeast reduction of one actin structure results in an increase on the other, suggesting, as pointed out by the reviewer, that pools compete for the available actin monomers. Translating this concept to our system implies that the lamellipodium and the central actin pool communicate to keep constant overall F-actin levels in DCs. This would mean that changes in a retracting protrusion affect F-actin levels at the central actin pool. In the unrevised version of our manuscript we first investigate this by observing LifeAct-eGFP expressing DCs migrating in PDMS pillar mazes, where splitting of the lamellipodium and consequent protrusion retraction happens frequently (**Extended Data Fig. 4e**). Indeed, in this setup lamellipodium retractions were accompanied by an increase of actin intensity at the central actin pool (**Extended Data Fig. 4f, g and Supplementary Video 12**), suggesting the existence of an equilibrium between F-actin at the cell front and the central pool. We agree with the reviewer that LifeAct-eGFP does not accurately reflect actin accumulation at protrusion regions, therefore, we turned to GFP-actin expressing DCs. Here we observed that actin intensity at protrusion sites is negatively correlated with actin intensity at the central actin pool (**Fig. 4g, h and Supplementary Video 12**), complementing our data on cells migrating in pillar mazes and strengthening our previous results (**Fig. 4i, Extended Data Fig. 4h, i and Supplementary Video 12**).

3. The cell fragmentation in complex environments is a striking phenotype. It is not obvious whether hyperprotrusion alone can explain it, or if it requires a change in cortical properties of the cell—for example the loss of cell coherence that accompanies myosin inhibition in some cells. On this note, it would be interesting to examine whether myosin activation is misregulated in the Dock8 null cells.

While we could not detect myosin at the central pool region (**Additional Data Fig. 2**, for details please refer to our reply to reviewer #1 point 3), we also find the resemblance between both phenotypes (myosin dysregulation and DOCK8 null cells) striking.

We have observed that DCs migrating through very narrow constrictions accumulate Myosin Light Chain (MLC) at the rear of the cell (**Additional Data Fig. 3a, b**). Perturbation of back contractility with the ROCK inhibitor Y-27632 (ROCK in.) substantially decreased the ability of cells to migrate through these narrow constrictions (**Additional Data Fig. 3c**). These results suggest that myosin contraction

in the back of DCs is essential to allow passage through very tight pores. Under this assumption, if DOCK8 depletion leads to myosin dysregulation, *DOCK8*^{-/-} DCs should display similar problems traversing narrow constricted channels. Interestingly, when tested in this set-up, we observed that *DOCK8* depleted cells were even faster than their WT control (**Additional Data Fig. 3d**). Moreover, when treated with Y-27632 (ROCK in.) *DOCK8* depleted cells lost the ability to transverse these narrow constrictions, mimicking the phenotype observed in WT DCs (**Additional Data Fig. 3e**). We are currently preparing a manuscript that describes the role of contractility in pore-passage in detail. These results not only suggest that *DOCK8* depleted cells are not affected by myosin dysregulation, but also show that the lack of the central actin pool in these cells confers an advantage when traversing narrow pores. Of note, this advantage (which we also observed in WASP deficient cells as show in Gaertner et al, 2022) only exists in environments where entanglement of the highly protrusive front that results from the excess of F-actin at the lamellipodium (**Fig. 3e and Fig. 4f**) is not possible.

Additional Data Figure 3. DOCK8 depleted cells show no indication of myosin dysregulation. **a.** Representative images of WT DCs expressing Myosin Light Chain-eGFP (MLC-eGFP – bottom, black) labeled with Hoechst (top, blue) migrating in constricted PDMS channels. Scale bar, 20 μm . **b.** Percentage of DCs displaying strong MLC-eGFP accumulation in the back of the cell while passing through either 2.5 or 1.2 μm constrictions. Note that we only observed significant MLC accumulation in the back of the cell in tight constrictions (1.2 μm). Violin plots show the median and quartiles. 2.5 μm constriction, n=139; 1.2 μm constriction, n=132. ** p-value=0.002525. Two-sided Mann-Whitney-Wilcoxon test. **c.** Cells migrating through 1.2 μm constricted channels upon treatment with DMSO (control) or Y-27632 compound (ROCK in., 10 μM). Passage of cells through the constrictions was manually curated and cells were classified into: “invading” (light blue) when successfully traversing the constrictions; “stuck” (blue), when stuck at the constriction; or “dead” (dark blue) when cell death was observed in the process of traversing the constriction. Data points correspond to the percentage of cells observed per technical replicate pooled from three independent experiments. DMSO treated cells, n=206; ROCK in. treated cells, n=113. **d.** Time that WT and DOCK8 depleted DCs spend to traverse 1.7 or 1.2 μm constrictions. Each data point corresponds to the time spent at the constriction for a single cell. Data was pooled from at least three independent experiments. Error bars show the SEM. 1.7 μm constriction: WT, n=97; DOCK8^{-/-}, n=65; 1.2 μm constriction: WT, n= 53; DOCK8^{-/-}, n=61. ** p-value=0.0010, *** p-value=0.0005. Unpaired Mann-Whitney test. **e.** Representative time-lapse images of WT and DOCK8^{-/-} DCs expressing LifeAct-eGFP (black, actin) and labeled with Hoechst (blue, nucleus) treated with DMSO (control) or Y-27632 compound (ROCK in, 10 μM) migrating in PMDS constricted channels. For all examples, from left to right: first image corresponds to the time point cells enter the straight channel (0 min); second image shows the time point cells reach the constriction, third image shows either a cell exiting or getting stuck at the constriction. Note that DOCK8 depleted cells lose the ability to transverse narrow constrictions upon Myosin II inhibition with Y-27632. Scale bar, 15 μm .

4. Is this central actin pool upon confinement unique to dendritic cells, or is it a more general feature of immune cells? From the author’s previous work, it seems less apparent in

lymphocytes, but it would be nice to have a direct comparison with other immune cells in the current manuscript. I'd be satisfied with just the LifeAct experiments.

We also found this structure in primary T cells, when they move under strong confinement and now added these data (**Supplementary Video 5 and Extended Data Fig. 1j, k**). We refrained from showing data of DOCK8 deficient T cells as another study by the lab of Judith Mandl that is currently being revised ([doi: https://doi.org/10.1101/2024.07.26.605273](https://doi.org/10.1101/2024.07.26.605273)) extensively characterizes the DOCK8-dependent central actin pool in T cells.

5. What is the relation between the central actin pool and bead displacement? For Fig 2c, it appears that the bulk of the bead displacement occurs before the central actin pool. Could the authors comment on this?

We thank the reviewer for this comment, as it made us realize that the example shown in Fig. 2 does not ideally represent the observed data. As the reviewer indicates and we acknowledge in the text, beads are first displaced by the periphery (front) of the cell. The important point here is that, statistically, bead displacement is significantly larger when the actin cloud passes under, even if for a few individual cells this increase is small. To improve clarity, we have changed the representative time trace in **Fig. 2c** of the revised version of our manuscript and have added more examples in **Extended Data Fig. 2 b**.

I would like to see the same plot in Dock8 null cells.

We apologize to the reviewer for not having included such plot in the original manuscript, as we agree that it can be very instructive. Following their suggestion, we have updated **Fig. 4 b-d** to resemble the data presented in **Fig. 2**. As a similar issue was also raised by reviewer #1 (point 3), please refer to or reply above for more details.

What about the rare cells where the central actin pool lags the nucleus? Or the subset of Cdc42 inhibited cells with and without a central actin pool? These experiments are important to establish the role of the central actin pool in vertical forces.

We agree on the importance of clearly distilling the role of the central actin pool in substrate deformations. While the usage of the Cdc42 inhibited cells would have been possible, we have doubts about the additional value of such experiments, as we think that the more extreme case of a complete absence of the central pool in DOCK8 knockout cells addresses the question more directly.

At the same time, we also find very interesting to understand and dissociate the contribution of the nucleus and central pool to substrate deformations, as the reviewer suggests. While we considered studying the very few cells that migrate nucleus-first in our assay, we decided to take a more direct approach and tested enucleated DCs (cytoplasts) instead. We observed that the central actin pool was present even in the absence of nucleus, and that it pushed upwards significantly more than the rest of the cell body, thus confirming its ability to deform the substrate independently from other cellular structures (**Fig. 2 f-h and Supplementary Video 6, Third part**).

Minor points

1. I don't see titles for the figures in the figure legends. Their addition would be useful to help understand what is in each.

We thank the reviewer for their suggestion and agree that titles in the figure legends improve the readability of the manuscript. We have added them to the revised version.

2. Now that there are much better tools, the authors shouldn't be using dominant negative Cdc42. Because this acts by sequestering upstream GEFs, any other GTPases that share this GEF will also be inhibited, and the literature abounds with artifacts from dominant negative GTPases. Knockdown or knockout of Cdc42 would be a much better approach.

Our lab has previously generated and studied conditional Cdc42^{-/-} DCs (Lammermann, Blood, 2009). Similar to DOCK8 depleted DCs, when moving in 2D substrates, Cdc42^{-/-} DCs often showed an

uncoordinated lamellipodium that resulted in less directional migration. However, in 3D collagen gels, these cells presented a rounded morphology and were completely unable to migrate, showing the importance of Cdc42 activity for efficient cell locomotion, especially in complex environments. In line with these results, to minimally perturb Cdc42 activity without compromising overall cell motility, we decided to transiently express the dominant-negative form of Cdc42 in our cells. As it was pointed out by the reviewer, there are plenty of examples of artifacts resulting from overexpressing dominant-negative Cdc42 as it can influence other GTPases. However, we believe it is informative that transient expression of this Cdc42 form can efficiently disrupt the central actin pool. In the revised version of this manuscript, we have moved the data from **Fig. 3** to **Extended Data Fig. 3i, j**. To support our observations, we decided to perturb Cdc42 activity with two different inhibitors that specifically target Cdc42: ZCL278 and ML141. ZCL278 directly interacts with Cdc42 at its Intersectin-1-binding site, possibly also interfering with GTP/GDP binding (Friesland et al, 2013). ML141 interaction with Cdc42, on the other hand, does not depend on the availability of a specific binding site. This molecule targets guanine-nucleotide bound Cdc42, induces ligand dissociation and locks Cdc42 in its inactive state (Hong et al, 2013). Treatment of DCs using both of these molecules resulted in a slight migration impairment in collagen gels (**Extended Data Fig. 3a, b**) and a decrease in the prevalence of the central actin pool to at least 60% of the cells (**Fig. 3a, b and Extended Data Fig. 3f, g**). Alltogether, these results show that specific perturbation of Cdc42 impacts actin polymerization at the central pool.

3. For cross-correlations, use seconds, not frames for the x-axis.

We agree with this suggestion and have edited all relevant x-axis accordingly.

4. The framing of mesenchymal vs amoeboid configuration is likely to be confusing to readers that may assume the cells are fully adopting a mesenchymal migration mode, which has many other features than just position of the nucleus. I'd recommend just talking about nuclear position instead of this mesenchymal framework.

We agree and in the revised version of the manuscript use the term “organelle re-orientation” instead of “amoeboid to mesenchymal transition”.

5. The authors propose that the central actin pool may protect the nucleus. Can they see evidence in changes in nuclear deformation (in agarose or collagen or microfluidic chambers) with and without the central actin pool?

DOCK8 deficient cells have indeed been shown to undergo massive nuclear deformations (Zhang et al, J Exp Med, 2014) suggesting that central actin might protect the nucleus. However, we assume that this possible function is hardly relevant for mature DCs, which are terminally differentiated cells that undergo programmed cell death within few days. Accordingly, mature DCs loose expression of proliferative factors and cannot undergo malignant transformation. A nucleo-protective function is more likely relevant in cells that can still proliferate. Indeed, a recent pre-print by the lab of Judith Mandl investigates this role of DOCK8 in T cells ([doi: https://doi.org/10.1101/2024.07.26.605273](https://doi.org/10.1101/2024.07.26.605273)). In this study, DOCK8 depleted T cells that lack the central actin pool show more double stranded DNA breaks than WT cells when migrating in collagen gels. We feel that it would not be fair to duplicate and publish this result in our manuscript.

6. Line 165: “WASP-dependent”

We have corrected it in the revised version.

7. Line 737: “MTOC-first” or “MTOC first”?; standardize hyphen usage throughout manuscript

We have standardized the hyphen usage throughout the manuscript by keeping “MTOC-first”.

8. Line 746: “DCs”

We have corrected it in the revised version of our manuscript.

9. Line 810: “AF555 (red)”

We have corrected it in the revised version of our manuscript.

10. Fig. 4g: Being able to see differences in speed very difficult to observe in Supp. Video 5 and only indicated by this cross-correlation plot – any way to depict these changes in speed in the video?

We agree that it is challenging to see changes in velocity in the representative video. To improve the understanding of our observations we added to the revised manuscript a plot depicting the changes of the central actin intensity, cell area and cell speed, of a single cell through time (**Extended Data Fig. 4h**).

11. Extended Data Fig. 4e: Background segmentation (“area where no sig. change in cell morph. was observed”, Line 973) seems to undergo large shape changes in panels and in Supp Vid 5; can the authors explain how background was chosen in more detail?

We apologize for this imprecision. We tried to clarify and substituted “area where no significant change in cell morphology was observed” by “area where no significant changes in actin intensity were observed”.